# All-optical reporting of inhibitory receptor driving force in the nervous system

Joshua S. Selfe ⓘ [1,2], Teresa J. S. Steyn[1,2], Eran F. Shorer[1,2,3], Richard J. Burman[4], Kira M. Düsterwald ⓘ [1,2,5], Ariel Z. Kraitzick ⓘ [1,2], Ahmed S. Abdelfattah ⓘ [6,7], Eric R. Schreiter ⓘ [8], Sarah E. Newey ⓘ [4], Colin J. Akerman ⓘ [4] & Joseph V. Raimondo ⓘ [1,2,9] ✉

Ionic driving forces provide the net electromotive force for ion movement across receptors, channels, and transporters, and are a fundamental property of all cells. In the nervous system, fast synaptic inhibition is mediated by chloride permeable GABA$_A$ and glycine receptors, and single-cell intracellular recordings have been the only method for estimating driving forces across these receptors (DF$_{GABAA}$). Here we present a tool for quantifying inhibitory receptor driving force named ORCHID: all-Optical Reporting of CHloride Ion Driving force. We demonstrate ORCHID's ability to provide accurate, high-throughput measurements of resting and dynamic DF$_{GABAA}$ from genetically targeted cell types over multiple timescales. ORCHID confirms theoretical predictions about the biophysical mechanisms that establish DF$_{GABAA}$, reveals differences in DF$_{GABAA}$ between neurons and astrocytes, and affords the first in vivo measurements of intact DF$_{GABAA}$. This work extends our understanding of inhibitory synaptic transmission and demonstrates the potential for all-optical methods to assess ionic driving forces.

Ionic driving forces represent the net electromotive force available for generating ion fluxes across receptors, channels, and transporters. A driving force is the product of both an ion's equilibrium potential and the potential difference across the membrane and constitutes the energy per unit charge that is available to mediate a host of fundamental cellular processes[1]. Despite their fundamental importance, the only way to estimate ionic driving forces has been using intracellular electrophysiological recording methods. This has limited measurements to small numbers of cells and, more fundamentally, in all cases, these methods either directly or indirectly affect the ion gradients that underlie the driving forces[2]. Meanwhile, methods for imaging intracellular ions cannot provide information on ionic driving force

because they are blind to the cell's membrane potential. Here we describe the first all-optical strategy for measuring ionic driving forces and demonstrate the utility of this approach for performing rapid, multi-cell measurements of undisturbed driving forces.

In the nervous system, fast synaptic inhibition is mediated by anion-permeable type A γ-aminobutyric acid receptors (GABA$_A$Rs) and glycine receptors (GlyRs). These ligand-gated receptors are primarily permeable to chloride (Cl$^-$) and, to a lesser extent, bicarbonate (HCO$_3^-$)[3]. The transmembrane gradients for Cl$^-$ and HCO$_3^-$ determine the inhibitory receptor reversal potential (E$_{GABAA}$) and, together with the membrane potential (V$_m$), set the ionic driving force across these receptors (e.g., DF$_{GABAA}$). Therefore, DF$_{GABAA}$, by controlling anion flux

[1]Division of Cell Biology, Department of Human Biology, University of Cape Town, Cape Town, South Africa. [2]Neuroscience Institute, University of Cape Town, Cape Town, South Africa. [3]Department of Neurology, School of Medicine, Johns Hopkins Hospital, Baltimore, Maryland, United States of America. [4]Department of Pharmacology, University of Oxford, Oxford, United Kingdom. [5]Gatsby Computational Neuroscience Unit, University College London, London, United Kingdom. [6]Department of Neuroscience, Brown University, Providence, Rhode Island, United States of America. [7]Carney Institute for Brain Science, Brown University, Providence, Rhode Island, United States of America. [8]Janelia Research Campus, Howard Hughes Medical Institute, Ashburn, Virginia, United States of America. [9]Wellcome Centre for Infectious Disease Research in Africa, Institute of Infectious Disease and Molecular Medicine, University of Cape Town, Cape Town, South Africa. ✉e-mail: joseph.raimondo@uct.ac.za

across inhibitory receptors, is a key factor in determining inhibitory synaptic signalling in the nervous system. As an exemplar of an all-optical strategy for measuring ionic driving forces, we present ORCHID (all-Optical Reporting of CHloride Ion Driving force) as a tool for quantifying $DF_{GABAA}$, which is based on combining genetically encoded voltage indicators with light-gated ion channels.

Intracellular $Cl^-$ concentration ($[Cl^-]_i$), and hence $E_{GABAA}$, is a dynamic variable that is known to differ between subcellular compartments, cell types, and over a range of timescales and network states[4,5]. Owing to the fact that $E_{GABAA}$ is typically close to $V_m$, relatively small changes to $E_{GABAA}$ or $V_m$ can have a large effect on the polarity and magnitude of $DF_{GABAA}$ and, thus, upon inhibitory signalling[6]. Alterations to neuronal $DF_{GABAA}$ have been implicated in multiple neurological diseases, including epilepsy, chronic pain, schizophrenia, and autism[7–10]. Similarly, $DF_{GABAA}$ is thought to be modulated during brain development as part of the GABAergic system's contribution to neural circuit formation[11,12]. In regions of the mature nervous system, it is believed that $DF_{GABAA}$ varies diurnally and is regulated as a function of wakefulness in neurons[13–15] and astrocytes[16]. Therefore, given the central importance of $DF_{GABAA}$ for both brain function and dysfunction, techniques for estimating $DF_{GABAA}$ are of major interest to the neuroscience field.

The most widely used electrophysiological strategy for estimating $DF_{GABAA}$ is gramicidin-perforated patch-clamp recording[2]. This technique provides electrical access to a target cell of interest without directly disturbing $[Cl^-]_i$. When combined with a means of generating ion selective conductances, such as activation of $GABA_ARs$ or light-sensitive channels with similar $Cl^-$ permeability[17], $E_{GABAA}$ and $DF_{GABAA}$ can be estimated. However, gramicidin recordings are technically challenging, have a low success rate and throughput, and perturb the intracellular levels of $HCO_3^-$, directly modifying $DF_{GABAA}$, as well as $K^+$ and $Na^+$, which can indirectly change $[Cl^-]_i$ and $DF_{GABAA}$ via the action of cation-$Cl^-$ cotransporters (CCCs) such as KCC2 and NKCC1[1]. Meanwhile, the best optical strategies for $[Cl^-]_i$ estimation include fluorescence lifetime imaging (FLIM) of the $Cl^-$-sensitive dye MQAE[18–20] and genetically encoded $Cl^-$ indicators[21]. Ratiometric genetically encoded $Cl^-$ indicators allow for high-throughput measurements and can be targeted to different cell types, although there are challenges associated with ion selectivity, signal-to-noise ratio, and measurements in intact systems[22–25]. More fundamentally, however, whilst currently available optical strategies can provide an estimate of $[Cl^-]_i$, they do not incorporate information about the cell's $V_m$ and, therefore, cannot quantify the force available in the transmembrane gradient for $Cl^-$ and $HCO_3^-$, which is $DF_{GABAA}$.

In this work, we show that ORCHID provides an easy-to-use, genetically encoded optical estimation of a cell's undisturbed $DF_{GABAA}$. ORCHID has been designed to incorporate the sensitive, chemigenetically encoded voltage indicator (GEVI), Voltron2-ST[26], with an independent, spectrally separate means of generating anion-selective currents by using light activation of *Guillardia theta* anion channelrhodopsin 2 (GtACR2)[27]. Voltron2-ST is therefore used to measure the magnitude and direction of GtACR2 anion-current-induced changes to $V_m$, thereby providing an estimate of $DF_{GABAA}$. We initially validate the ORCHID approach in mice by demonstrating that it replicates estimates of $DF_{GABAA}$ that rely upon the activation of endogenous $GABA_AR$ conductances. ORCHID is shown to provide reliable measurements of resting and dynamic $DF_{GABAA}$ in different genetically targeted cell populations and across a timescale of seconds and hours. Furthermore, ORCHID confirms theoretical predictions regarding the biophysical mechanisms that establish $DF_{GABAA}$. Finally, we use ORCHID to reveal cell-type-specific differences in activity-dependent $DF_{GABAA}$ changes during periods of network activity and to provide the first in vivo measurements of intact $DF_{GABAA}$ in mouse cortical neurons. Together, our results demonstrate that ORCHID allows high-throughput, dynamic estimation of $DF_{GABAA}$ in a cell-type-specific manner. This will assist in the study of inhibitory synaptic transmission and $Cl^-$ homoeostasis in the nervous system and establish a precedent for using all-optical methods to assess ionic driving force more generally.

## Results

### ORCHID provides an optical estimate of the magnitude and direction of inhibitory receptor driving force

In order to optically measure $DF_{GABAA}$, we designed a strategy based on combining a genetically encoded optical reporter of $V_m$, with the simultaneous activation of a light-sensitive transmembrane anion channel (Fig. 1a). This provides a measurement of the difference between the $V_m$ and $E_{GABAA}$ (i.e., $DF_{GABAA}$), which can vary both in magnitude and polarity (Fig. 1b). The resulting ORCHID strategy utilises the recently developed soma-targeted GEVI Voltron2 (Voltron2-ST), and the light-activated anion channel GtACR2, separated via a P2A linker sequence for expression under the same promoter and translation of two separate proteins (Fig. 1c). The ORCHID construct has a double-floxed inverted-orientation (DIO) design for targeting Cre-expressing cells and is under the control of the EF1α promoter (Supplementary Fig. 1). Cre-expressing neurons in mouse hippocampal organotypic slices were successfully transduced with adeno-associated virus (AAV) vectors encoding ORCHID ("Methods"). Voltron2-ST works by irreversibly binding, via a HaloTag protein, one of a number of different fluorescent Janelia Fluor (JF) HaloTag ligands (dyes), each of which has different spectral properties[28]. Prior to imaging cells expressing ORCHID, a dye incubation step was carried out with a dye that was selected for having spectral properties complementary to the experimental paradigm.

For ORCHID to work, it is critical to achieve spectral separation of the voltage readout using Voltron2-ST, from the manipulation of anion currents using GtACR2. Whole-cell patch-clamp recordings from pyramidal neurons expressing GtACR2 in hippocampal organotypic slices revealed a wide activation spectrum, with currents being generated by blue (475/28 nm), green (555/28 nm), and yellow (575/25 nm) excitation wavelengths, but not by orange (599/13 nm) or red (635/22 nm) wavelengths (Fig. 1d). Therefore, we utilised the HaloTag dye $JF_{608}$ together with continuous orange (599/13 nm) excitation light to ensure that Voltron2-ST imaging elicited no GtACR2 currents (Fig. 1e). This steady-state illumination produces negligible photocurrents by Voltron2-ST itself[26]. Activation of GtACR2 with blue light (475/28 nm) does not significantly excite $JF_{608}$ fluorescence[29]; nonetheless, blue light delivery was rapidly interleaved with camera acquisition to ensure that blue-light-generated autofluorescence would not cause artefacts in the imaging channel for Voltron2-ST labelled with $JF_{608}$ (Voltron2$_{608}$-ST) ("Methods"). The GtACR2 stimulation time was chosen to optimise the signal-to-noise ratio (SNR) in recordings ("Methods"). Imaging neurons expressing only Voltron2$_{608}$-ST confirmed that there was no change in fluorescence during blue light delivery (Fig. 1f). Voltron2-ST fluorescence readout ($\Delta F/F_0$) was plotted as a negative for intuitive understanding of the underlying $V_m$ shifts, as Voltron2-ST fluorescence increases with a decrease in $V_m$.

Whole-cell patch-clamp recordings from ORCHID-expressing neurons confirmed that Voltron2$_{608}$-ST reports linear fluorescence changes in response to voltage steps, thereby establishing that $\Delta F/F_0$ can be used to estimate changes in voltage ($\Delta V$). This was used to estimate $DF_{GABAA}$ in subsequent analyses, including where cells were imaged alone (Fig. 1g–j; "Methods"). In ORCHID-expressing neurons, blue light stimulation elicited clear and reproducible $V_m$ shifts, which showed reliable reproducibility within the same neuron (Fig. 1k, l and Supplementary Table 1).

To assess ORCHID's ability to quantify $DF_{GABAA}$, we performed three variations of patch-clamp recordings with simultaneous voltage imaging in ORCHID-expressing neurons. Firstly, we used current-clamp recordings with a standard low-$Cl^-$ internal solution to impose different $DF_{GABAA}$ values in ORCHID-expressing neurons by injecting

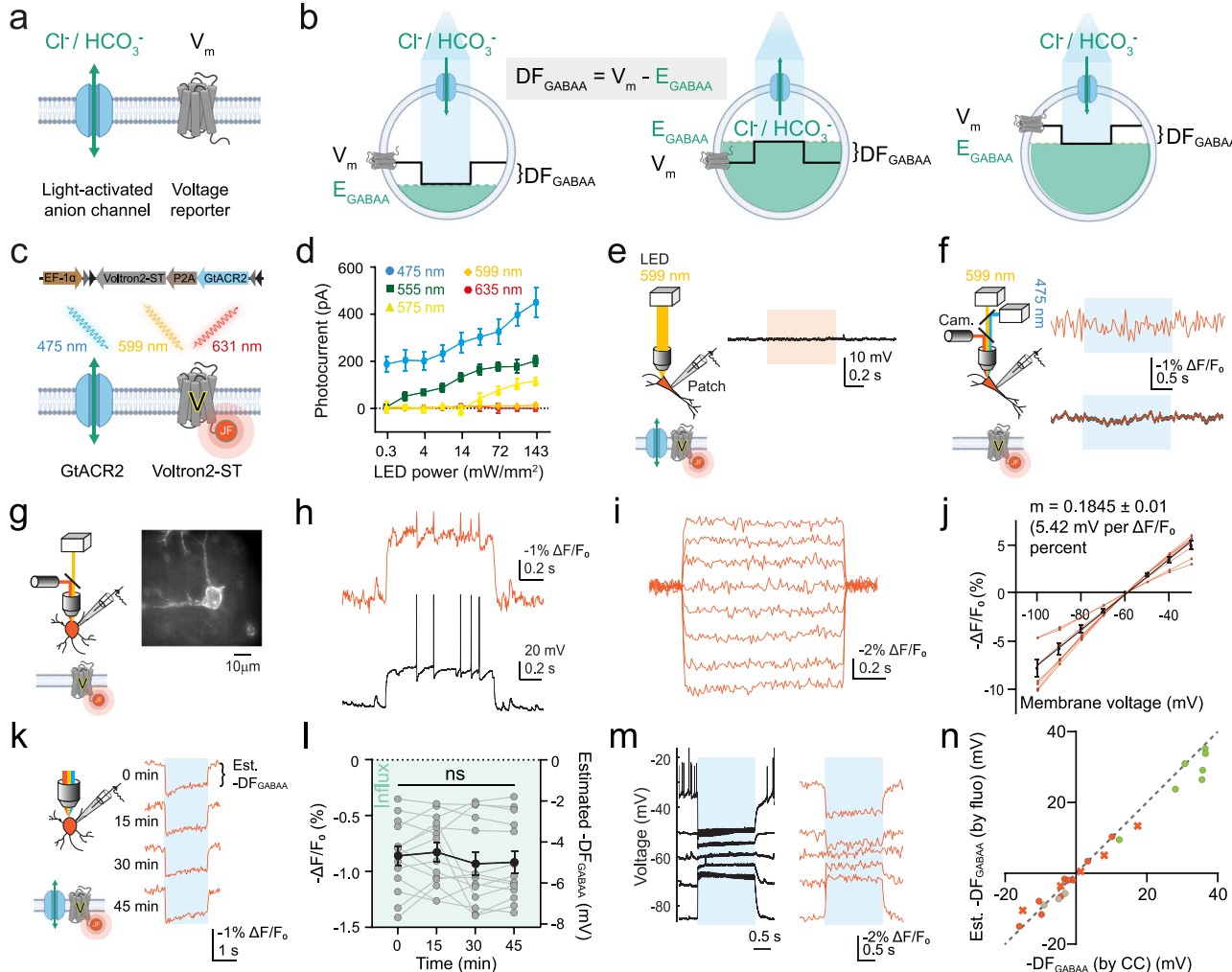

**Fig. 1 | ORCHID provides an optical estimate of the magnitude and direction of inhibitory receptor driving force. a** Conceptual design of an all-optical strategy to estimate $DF_{GABAA}$, combining a light-activated anion channel with an optical voltage reporter. **b** Activation of an anion conductance (blue) drives $V_m$ (black trace) toward $E_{GABAA}$ (green dotted line). The $V_m$ shift upon anion channel activation provides an estimate of $DF_{GABAA}$ ($DF_{GABAA} = V_m - E_{GABAA}$). Left, $E_{GABAA}$ is negative compared to $V_m$, constituting a driving force that results in anion influx and hyperpolarization. Middle, due to higher $[Cl^-]_i$ (green shading), $E_{GABAA}$ is positive compared to $V_m$, and $DF_{GABAA}$ causes anion efflux and depolarisation. Right, despite identical $E_{GABAA}$, different $V_m$ results in a $DF_{GABAA}$ that causes anion influx and hyperpolarization. **c** ORCHID utilises co-expression of GtACR2 with Voltron2-ST labelled with $JF_{608}$ HaloTag dye. **d** Whole-cell patch-clamp recorded GtACR2 photocurrents at different wavelengths from organotypic hippocampal neurons (475 nm, $n = 7$ cells; 555 nm, $n = 7$ cells; 575 nm, $n = 6$ cells; 599 nm, $n = 9$ cells; 635 nm, $n = 3$ cells). Data are presented as mean values ± SEM. **e** Current-clamp recording (patch) from a pyramidal neuron expressing ORCHID demonstrates no change in $V_m$ during 1 s 599 nm LED light exposure (orange rectangle). **f** Voltage imaging (sCMOS camera, i.e., cam.) of an organotypic hippocampal pyramidal neuron expressing Voltron2$_{608}$-ST alone reveals that blue light delivery strobed between camera exposures for 2 s ("Methods") causes no change in measured fluorescence in either a recording from a single cell (top, orange trace), or in the averaged recorded fluorescence from 29 neurons (mean, orange trace ± SEM, grey shading). **g** Setup for patch-clamp characterisation of Voltron2$_{608}$-ST. Inset: the widefield image of the interneuron patched in (**h**), and part of the dataset in (**j**); scale bar: 10 μm. **h** Current-clamp recording (black trace) and Voltron2$_{608}$-ST fluorescence response (orange trace) following a 200 pA current injection. **i** Voltron2$_{608}$-ST fluorescence response to 10 mV voltage

steps ($V_{hold} = -60$ mV). **j** Fluorescence-voltage relationship of Voltron2$_{608}$-ST. Linear regression results in a slope of 0.1845 ± 0.01, which equates to 5.42 mV per $\Delta F/F_0$ percent ($R^2 = 0.9311$, $n = 7$ cells). Orange traces are recordings from individual neurons, and the black trace is the linear regression of the data, with error bars showing SEM. **k** $DF_{GABAA}$ estimated (est.) using fluorescence transients (orange traces) of an ORCHID-expressing interneuron following 2 s blue light activation (blue rectangle) was stable over 45 min. **l** Population data showing stable estimates of $DF_{GABAA}$ using ORCHID (repeated measures one-way ANOVA, $P = 0.2469$, $n = 12$ cells). Tukey's multiple comparison tests were used post-hoc and found no statistical difference between any pair of means (Supplementary Table 1). Green shading indicates the direction of anion flux. Here and henceforth, grey data points indicate single recordings from individual cells, while black data points indicate the mean with error bars indicating SEM. **m** ORCHID used to estimate $DF_{GABAA}$ in a neuron current clamped at a range of $V_m$ values (left panel, $V_m$; right panel, fluorescence). Action potentials are truncated due to averaging. **n** The fluorescence-voltage relationship of Voltron2$_{608}$-ST (**j**) was used to convert $\Delta F/F_0$ measurements at each initial $V_m$ to estimated (est.) $DF_{GABAA}$ values (by fluorescence, i.e., fluo). This was done for neurons patched with low-$Cl^-$ internal solution (orange; crosses from cell in (**m**); $n = 14$ recordings from 7 cells), neurons patched with gramicidin perforation (beige; $n = 3$ recordings from 3 cells), and neurons patched with high-$Cl^-$ internal solution (green; $n = 7$ recordings from 7 cells). Where available for low-$Cl^-$ internal recordings, holding potentials below, above, and close to $E_{GABAA}$ are plotted for each cell. These optically estimated $DF_{GABAA}$ values did not differ from $DF_{GABAA}$ recorded using current-clamp (CC) recordings ($R^2 = 0.9801$; Runs test, one-tailed, deviation from linearity: $P = 0.9360$). Dashed-grey identity line. ns = not significant ($P > 0.05$); error bars indicate mean ± SEM.

current via the patch pipette. Meanwhile, voltage imaging of ORCHID was performed under orange light, and anion currents were generated via blue-light activation of ORCHID (Fig. 1m, n). The relationship between $\Delta F/F_0$ and $\Delta V$ (Fig. 1j) was used to convert the measured $\Delta F/F_0$ values into estimations of anion-current-induced changes to $V_m$[30]; when analysing recordings from multiple neurons, these optically determined $DF_{GABAA}$ values were found to be equivalent to $DF_{GABAA}$ measured from the simultaneous current-clamp recordings (Fig. 1n; $R^2 = 0.9801$; Runs test, one-tailed, deviation from linearity: $P = 0.9360$, $n = 24$ recordings from 17 cells). This demonstrated that the ability of ORCHID to estimate $DF_{GABAA}$ is as accurate as using current-clamp recordings, with both methods accurately determining the direction of $DF_{GABAA}$ and inclined to underestimate the full magnitude of $DF_{GABAA}$. Secondly, we used gramicidin-perforated patch-clamp recordings to quantify the similarity between $DF_{GABAA}$ estimated optically using ORCHID and $DF_{GABAA}$ measured through a perforated patch-clamp recording with undisturbed $[Cl^-]_i$ (Fig. 1n and Supplementary Fig. 2a). Measurements of $DF_{GABAA}$ using these two techniques were consistent with one another (Fig. 1n and Supplementary Fig. 2b, Mann-Whitney test, two-tailed, $P = 0.7969$). Finally, ORCHID's ability to detect a change in $DF_{GABAA}$ was further confirmed by comparing measurements of $DF_{GABAA}$ made before and after a neuron experienced an increase in $[Cl^-]_i$ imposed by patching with a high $Cl^-$ internal solution (Fig. 1n and Supplementary Fig. 3).

As further validation, we assessed ORCHID's ability to accurately determine $DF_{GABAA}$ as compared to when anion currents were generated by activating endogenous $GABA_A$Rs in different neuronal populations in hippocampal organotypic slices. Transgenic mice expressing Cre recombinase in different cell types were used to restrict reporter expression to either CaMKIIα+ pyramidal neurons or GAD2+ interneurons. Voltron2-ST labelled with JF$_{549}$ (Voltron2$_{549}$-ST) was used to assess $DF_{GABAA}$ via endogenous $GABA_A$Rs on the cell soma, which were activated by picolitre delivery of GABA (500 μM) via a glass micropipette and in the presence of a $GABA_B$R antagonist (CGP-55845; 5 μM). This was found to be an effective method of estimating $DF_{GABAA}$ (Supplementary Figs. 4–6 and Supplementary Table 2) and revealed that both CaMKIIα+ pyramidal neurons and GAD2+ interneurons exhibit hyperpolarizing $DF_{GABAA}$ under these conditions, with GAD2+ interneurons exhibiting slightly greater $DF_{GABAA}$ (Fig. 2a–c). When these experiments were conducted using ORCHID, our all-optical strategy revealed similar hyperpolarizing $DF_{GABAA}$ values for both neuronal populations, again with slightly greater $DF_{GABAA}$ in GAD2+ interneurons compared to CaMKIIα+ pyramidal neurons (Fig. 2d–f). $DF_{GABAA}$ estimates made using GtACR2 versus endogenous $GABA_A$Rs were not statistically distinguishable (Supplementary Fig. 7; CaMKIIα+ pyramidal neurons: Mann-Whitney test, two-tailed, $P = 0.0660$; GAD2+ interneurons: Mann-Whitney test, two-tailed, $P = 0.1819$).

## ORCHID strategies confirm theoretical predictions regarding inhibitory receptor driving force

The accepted view is that $DF_{GABAA}$ is predominantly established by the action of CCCs such as KCC2 and NKCC1, which are able to use secondary active transport to move $Cl^-$ against its transmembrane concentration gradient by utilising cation gradients established by the primary active transporter, $Na^+/K^+$-ATPase[1]. However, work using the genetically encoded $Cl^-$ indicator, Clomeleon, challenged this idea by proposing that impermeant anions (and not CCCs) set $[Cl^-]_i$, $E_{GABAA}$, and $DF_{GABAA}$[31]. We sought to examine these biophysical principles by first using a theoretical model to make predictions about the mechanisms that establish $DF_{GABAA}$. We extended a computational biophysical model of a single neuron based on the pump-leak mechanism that incorporated the permeant ions ($Na^+$, $K^+$, $Cl^-$, $HCO_3^-$) and their movement via leak channels, receptors, the $Na^+/K^+$-ATPase, and the major $Cl^-$ extruder in mature neurons, KCC2. The model also incorporated impermeant anions, X, with average charge $z$[32], pH

control via the $HCO_3^-$-buffering system, plus $Na^+/H^+$ exchange (NHE), dynamic volume, water permeability, and synaptic $GABA_A$Rs that were able to elicit $Cl^-$ and $HCO_3^-$ conductances and resulting changes in $V_m$ (Fig. 3a). As expected, simulating a block of KCC2 (i.e., a decrease in the conductance of KCC2 [$g_{KCC2}$]) in the model caused a positive shift in $E_{Cl}$, $E_{GABAA}$, and $[Cl^-]_i$, with a small effect on $V_m$ (Fig. 3b). This resulted in a substantial shift in $DF_{GABAA}$, causing a decrease in amplitude and an inversion in polarity of $GABA_A$R-mediated potentials (Fig. 3c). Meanwhile, simulating the addition of impermeant anions by reducing their average charge ($z$) generated a persistent negative shift in $E_{Cl}$, $E_{GABAA}$, and $[Cl^-]_i$, consistent with previous observations[32]. However, this manipulation caused ionic redistribution such that $V_m$ also shifted and in a manner proportional to the effects upon $E_{Cl}$ and $E_{GABAA}$ (Fig. 3d). Therefore, $GABA_A$R-mediated potentials were of a similar size following the addition of impermeant anions (Fig. 3e).

To empirically test these predictions, we used ORCHID strategies (i.e., activation of $GABA_A$Rs or stimulation of GtACR2) to measure the effect of manipulations upon $DF_{GABAA}$ in hippocampal organotypic CaMKIIα+ pyramidal neurons. First, we observed that the addition of the selective inhibitor of KCC2, VU0463271 (VU; 10 μM), caused a robust depolarising shift in estimated $DF_{GABAA}$ (Fig. 3f–i). This was evident when $DF_{GABAA}$ was measured either via activation of $GABA_A$Rs (Fig. 3f, g) or all-optically (Fig. 3h, i), and the effect upon $DF_{GABAA}$ reversed following washout of VU (Fig. 3f–i). Second, we assessed $DF_{GABAA}$ before and after the addition of intracellular impermeant anions, which was achieved by single-cell electroporation of fluorescently tagged anionic dextrans[32] (Alexa Fluor 488) ("Methods") (Fig. 3j). Successful addition of impermeant anions was confirmed by directly observing the fluorescent dextrans within the neuron of interest, and $DF_{GABAA}$ was estimated from the size and polarity of $GABA_A$R responses before and after the electroporation procedure (Fig. 3k, l). Consistent with the prediction from the modelling work, the addition of impermeant anions had no detectable effect upon the experimentally determined $DF_{GABAA}$ (Fig. 3m). Therefore, ORCHID strategies are able to confirm theoretical predictions about the mechanisms that establish $DF_{GABAA}$.

## ORCHID reveals dynamic ion driving forces over different timescales

In neurons, $[Cl^-]_i$ and $E_{GABAA}$ are considered dynamic variables under the control of multiple activity-dependent processes and are thought to vary over a range of timescales[4]. Therefore, we sought to use ORCHID to investigate $DF_{GABAA}$ dynamics and first examined periods of elevated network activity, during which intense activation of $GABA_A$Rs is hypothesised to cause transient changes in $DF_{GABAA}$[6]. ORCHID was expressed in CaMKIIα+ pyramidal neurons or GAD2+ interneurons, and we used low-$Mg^{2+}$ artificial cerebrospinal fluid (aCSF) to induce seizure-like events (SLEs) in mouse hippocampal organotypic brain slices. $DF_{GABAA}$ was estimated using ORCHID in target cells, whilst a simultaneous whole-cell current-clamp recording from a nearby CA1/CA3 pyramidal neuron provided an independent readout of network activity (Fig. 4a–d). The SLEs were characterised by pronounced depolarising shifts in $V_m$ and intense action potential firing, which could be observed both in the current-clamp recordings and in ORCHID's voltage signals (Fig. 4a, c). CaMKIIα+ pyramidal neurons showed substantial activity-dependent changes in $DF_{GABAA}$, exhibiting shifts in polarity from a hyperpolarizing $DF_{GABAA}$ at baseline to a depolarising $DF_{GABAA}$ immediately post-SLE, which then returned to a hyperpolarizing $DF_{GABAA}$ > 2 min after the SLE (Fig. 4b). Similarly, GAD2+ interneurons exhibited a hyperpolarizing $DF_{GABAA}$ at baseline that flipped polarity to become a depolarising $DF_{GABAA}$ immediately post-SLE, and subsequently returned to a hyperpolarizing $DF_{GABAA}$ after > 2 min (Fig. 4d). These dynamics were confirmed by using endogenous $GABA_A$R activation to estimate $DF_{GABAA}$ in CaMKIIα+ pyramidal neurons and GAD2+ interneurons (Supplementary Fig. 8).

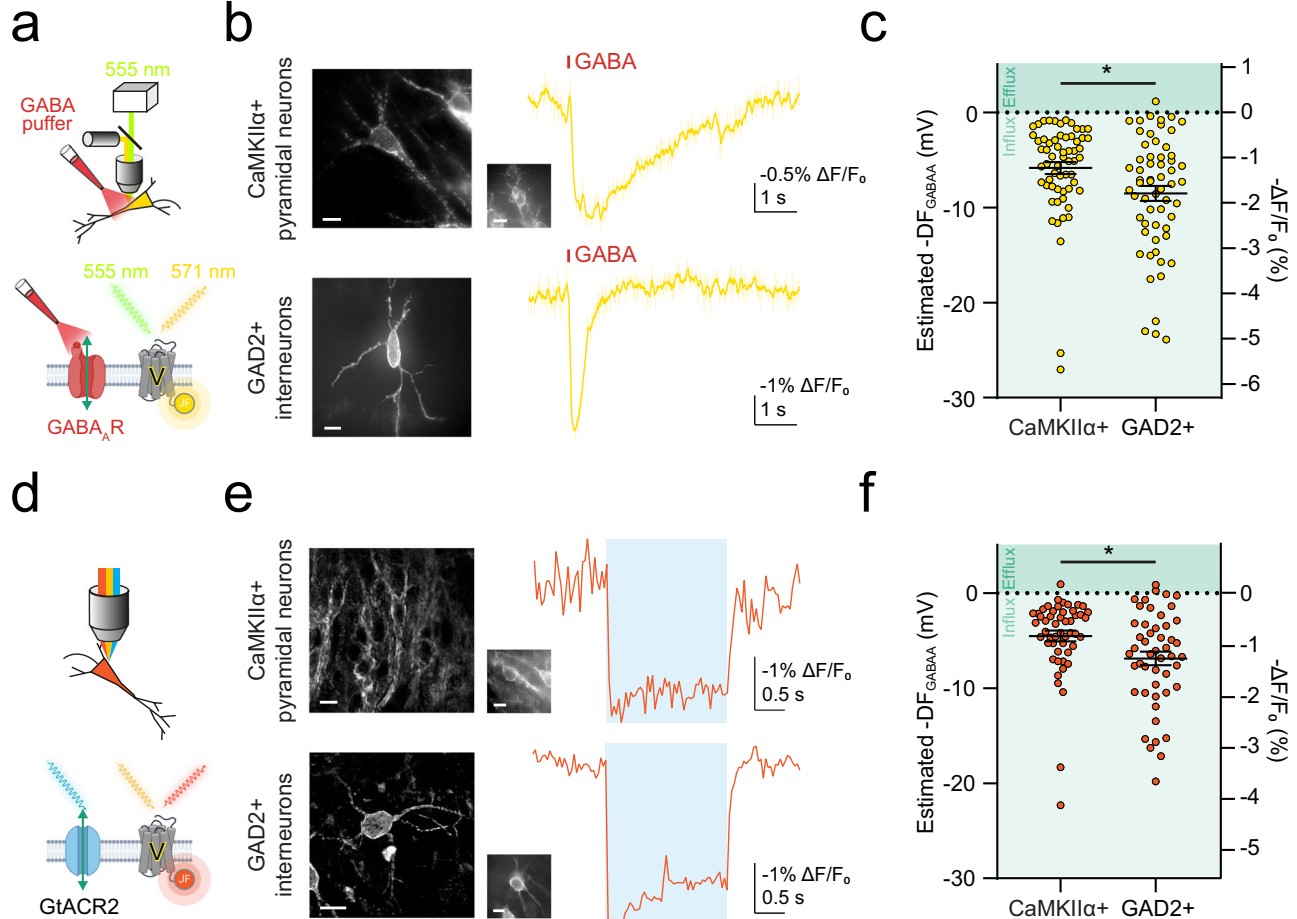

**Fig. 2 | ORCHID reports endogenous inhibitory receptor driving force in different neuronal populations. a** Schematics showing $DF_{GABAA}$ estimation by imaging Voltron2$_{549}$-ST together with activation of endogenously expressed GABA$_A$Rs via 100 ms soma-directed GABA application (puffer, red; 500 μM). CGP-55845 (5 μM) was used to block GABA$_B$Rs. **b** Widefield fluorescence images of CaMKIIα+ pyramidal neurons (top) and GAD2+ interneurons (bottom) expressing Voltron2$_{549}$-ST (left), and fluorescence recordings used to estimate $DF_{GABAA}$ from each cell type (right, yellow traces) in hippocampal organotypic slices, with recorded cells shown via inset if not in the main image; scale bar: 10 μm. The red bar indicates GABA application. Raw trace is shown in light yellow, and smoothed trace (window length = 7) is overlayed in darker yellow. **c** Population data showing estimated $DF_{GABAA}$ was different between cell types (CaMKIIα+: 5.85 ± 0.62 mV vs

GAD2+: 8.52 ± 0.79 mV, Mann-Whitney test, two-tailed, $P = 0.0068$, $n = 62$ and 60 cells respectively). Green shading indicates the direction of anion flux. **d** Schematics depicting ORCHID's use for estimating $DF_{GABAA}$ in neurons. **e** Left, confocal images of CaMKIIα+ pyramidal neurons (top) and GAD2+ interneurons (bottom) expressing ORCHID in mouse hippocampal organotypic brain slices. Right, fluorescence recordings were used to estimate $DF_{GABAA}$ from each cell type, with widefield images of the recorded cells inset; scale bar: 10 μm. **f** Estimated $DF_{GABAA}$ was significantly different between cell types (CaMKIIα+: −4.52 ± 0.57 mV vs GAD2+: −6.88 ± 0.72 mV, Mann-Whitney test, two-tailed, $P = 0.0057$, $n = 51$ and 49 cells respectively). -$DF_{GABAA}$ values plotted. ns = not significant ($P > 0.05$); *$P ≤ 0.05$; error bars indicate mean ± SEM.

Furthermore, shifts in $DF_{GABAA}$ have been postulated to occur immediately prior to seizure onset and to form part of the mechanism by which seizures initiate[33]. In line with this, ORCHID detected more modest decreases in the magnitude of $DF_{GABAA}$ between baseline and immediately prior to the SLE (< 15 s prior to SLE onset) in both CaMKIIα+ pyramidal neurons and GAD2+ interneurons (Fig. 4b, d), suggesting that $DF_{GABAA}$ begins to break down before SLE onset (Supplementary Fig. 9). Interestingly, GtACR2 activation during ORCHID recordings from GAD2+ interneurons elicited further network bursts following an SLE (Fig. 4c). This occurred in recordings from 8/12 GAD2+ interneurons, and not when recording from CaMKIIα+ pyramidal neurons (0/15), highlighting the network effects of activity-induced intracellular Cl⁻ accumulation in interneurons (Supplementary Fig. 10).

Having demonstrated ORCHID's ability to report $DF_{GABAA}$ dynamics on a timescale of seconds, we next sought to assess ORCHID's potential to measure longer-term changes in a neuron's $DF_{GABAA}$, such as those that have been linked to activity-dependent changes in CCCs[34]. To this end, ORCHID was used to track $DF_{GABAA}$ in the same

GAD2+ interneurons before and after a 180 min treatment with either 4-aminopyridine (4-AP; 50 μM) to increase neuronal activity or tetrodotoxin (TTX; 1 μM) to silence neuronal activity (Fig. 4e). When $DF_{GABAA}$ measurements were made after the 180 min treatment and a subsequent 10 min incubation in TTX, ORCHID revealed that raising neuronal activity had caused a sustained and robust depolarising shift in neuronal $DF_{GABAA}$ (Fig. 4f, g), which was not evident in neurons that had been silenced with TTX (Fig. 4h, i), or in neurons exposed to vehicle control for the equivalent 180 min period (Supplementary Fig. 11). Hence, ORCHID can provide high-throughput measurements of resting and dynamic $DF_{GABAA}$ across timescales of seconds and hours.

## Astrocytes sustain an outward anion driving force during enhanced network activity

Astrocytes are known to play important roles in regulating and maintaining neuronal ion gradients by modulating extracellular ion concentrations. However, due to the difficulty in performing

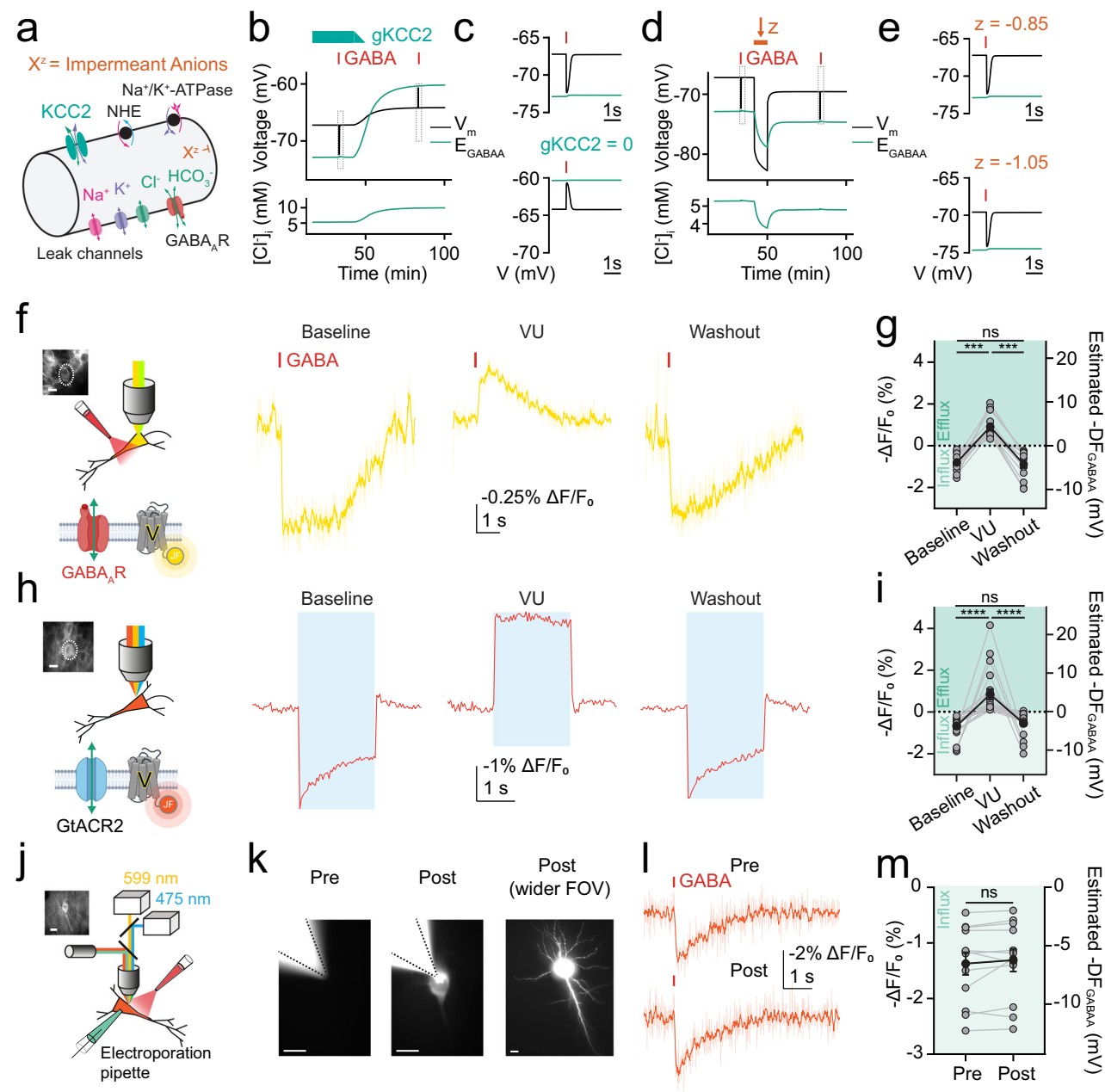

electrophysiological recordings from this cell type, it has not previously been possible to measure astrocytic $DF_{GABAA}$ across different types of network activity, although $[Cl^-]_i$ has been imaged in astrocytes before[16,35]. Using the Cre-lox system, we restricted ORCHID expression to GFAP+ astrocytes in mouse hippocampal organotypic slices and optically estimated $DF_{GABAA}$. We observed that, in stark contrast to neurons, the astrocytic $DF_{GABAA}$ under baseline conditions was strongly depolarising, consistent with previous reports[36] (Fig. 5a–c). A related idea that has been proposed is that astrocytes may serve to continuously supplement anions into the extracellular space during periods of enhanced network activity, and thereby help to maintain the effectiveness of $GABA_AR$-mediated inhibitory synaptic transmission[16,35]. To test this explicitly, we used ORCHID to continuously monitor astrocytic $DF_{GABAA}$ across periods of intense network activity, which was achieved by inducing SLEs through low-$Mg^{2+}$ aCSF. Whilst there was a modest decrease in the magnitude of $DF_{GABAA}$ during the SLE, ORCHID revealed that the astrocytes continued to exhibit a strongly depolarising $DF_{GABAA}$ before and after the SLE, and

are therefore capable of maintaining a driving force for anion efflux during substantial changes in network activity (Fig. 5d, e and Supplementary Fig. 12). The large increase in fluorescence at the initiation of the SLE is likely due to a large depolarisation in the astrocytic $V_m$. When compared to neurons, markedly different resting and dynamic $DF_{GABAA}$ in astrocytes suggest that they differ in how they utilise major $Cl^-$ regulatory mechanisms such as NKCC1 and KCC2, which act to transport $Cl^-$ in and out of the cell, respectively. To assess this, we performed single-nucleus RNA sequencing (snRNAseq) on the same hippocampal brain slice preparations ("Methods") and distinguished pyramidal neurons and astrocytes using unbiased clustering of the transcriptomic profiles (Fig. 5f). After normalising for total gene expression, we observed that KCC2 (*Slc12a5*) was more highly expressed in pyramidal neurons than astrocytes. Meanwhile, NKCC1 (*Slc12a2*), although comparatively low in both cell types, exhibited higher expression in astrocytes than pyramidal neurons (Fig. 5f). This confirms the established idea that KCC2 is highly expressed in mature neurons, and NKCC1 is expressed at a higher level than KCC2 in

**Fig. 3 | ORCHID strategies confirm theoretical predictions regarding inhibitory receptor driving force. a** Biophysical computational model of ion dynamics based on the pump-leak mechanism. Transmembrane movement of permeant ions ($Na^+$, $K^+$, $Cl^-$, $HCO_3^-$) was modelled though leak channels, KCC2, $Na^+$/$K^+$-ATPase, $Na^+$/$H^+$ exchanger (NHE), and $GABA_A$Rs. Impermeant anions, X, with average charge $z$ could not cross the membrane. **b** Top, simulated decrease in the conductance of KCC2 ($g_{KCC2}$) from $20\,\mu S/cm^2$ to $0\,\mu S/cm^2$ between 41 min and 50 min (teal bar) caused an increase in $E_{GABAA}$ (green) and smaller changes in $V_m$ (black). Simulated synaptic $GABA_A$R activation (red bar and grey rectangle) delivered before and after simulated KCC2 blockade. Bottom, increased $[Cl^-]_i$ following $g_{KCC2}$ reduction. **c** Enlarged view of grey rectangles in (**b**) shows simulated synaptic $GABA_A$R activation (red bar) in the neuron causes hyperpolarization with KCC2 intact (top), but depolarisation with KCC2 blocked (bottom). **d** As in (**b**) but with $z$ shifted from $-0.85$ to $-1.05$ between 41 min and 50 min (orange bar). Top, proportional drop in $V_m$ (black) and $E_{GABAA}$ (green) reflected a relatively unchanged $DF_{GABAA}$ despite altered $z$. Bottom, reduced $[Cl^-]_i$ in response to decreased $z$. **e** Enlarged view of grey rectangles in (**d**) shows simulated synaptic $GABA_A$R activation results in similarly sized hyperpolarization in the neuron with default $z$ of $-0.85$ (top), and with reduced $z$ of $-1.05$ (bottom). **f** Voltage imaging (Voltron2$_{549}$-ST, yellow traces) of a CaMKIIα+ pyramidal neuron in a hippocampal organotypic slice (inset, scale bar: $10\,\mu m$; white dotted line indicates cell from which recording was made) following activation of endogenous $GABA_A$Rs at baseline (left), during KCC2 blockade by VU (middle; $10\,\mu M$), and after VU washout (right). Raw traces are shown in light yellow, and smoothed traces (window length = 7) are overlayed in darker yellow. **g** Population data showing blockade of KCC2 by VU caused a significant depolarisation of $DF_{GABAA}$ in CaMKIIα+ neurons (left, Wilcoxon matched-pairs signed rank tests, two-tailed, wash-in $P = 0.0001$, washout $P = 0.0001$, $n = 14$ cells, baseline vs washout comparison, $P = 0.6698$). **h** All-optical ORCHID was used to record $DF_{GABAA}$ as in (**f**). **i** Population data showing that VU caused a significant shift in $DF_{GABAA}$ recorded using ORCHID in CaMKIIα+ pyramidal neurons (left, Wilcoxon matched-pairs signed rank tests, two-tailed, wash-in $P = 0.000002$, washout $P = 0.000002$, $n = 20$ cells, baseline vs washout comparison, $P = 0.1536$). **j** Schematic of the experimental setup for electroporation of CaMKIIα+ pyramidal neurons with fluorescently tagged (Alexa Fluor 488) anionic dextran. Inset: widefield fluorescence image of Voltron2$_{608}$-ST fluorescence of targeted neuron shown in (**k**) and (**l**), and part of the dataset in (**m**); scale bar: $10\,\mu m$. **k** Successful electroporation of neurons with anionic dextran verified by observing fluorescence restricted to the cell of interest; scale bars: $10\,\mu m$. **l** Activation of endogenous $GABA_A$Rs during Voltron2$_{608}$-ST voltage imaging was used to estimate $DF_{GABAA}$ from the pyramidal neuron in (**j**) and (**k**), both pre-electroporation (pre) and post-electroporation (post). Raw trace shown in light orange, and smoothed trace (window length = 7) overlayed in darker orange. **m** Population data showing that the addition of impermeant anions had no effect on estimated $DF_{GABAA}$ (paired $t$ test, two-tailed, $P = 0.2506$, $n = 12$ cells). ns = not significant ($P > 0.05$); *$P \leq 0.05$; ***$P \leq 0.001$; ****$P \leq 0.0001$; error bars indicate mean ± SEM.

---

astrocytes[37–39]. Thus, ORCHID reveals unique features of astrocytic $DF_{GABAA}$ compared to neurons, that correlate with differences in their $Cl^-$-transport mechanisms.

### In vivo determination of resting and dynamic inhibitory receptor driving forces in neurons

Finally, ORCHID offers the opportunity to generate estimates of undisturbed $DF_{GABAA}$ in the intact brain. We therefore assessed ORCHID's ability to report resting and dynamic $DF_{GABAA}$ in vivo. ORCHID was expressed in layer 1 (L1) GAD2+ interneurons in the mouse primary somatosensory cortex (S1), and anaesthetised in vivo imaging was combined with simultaneous local-field potential (LFP) recordings to provide an independent measure of local population activity (Fig. 6a; "Methods"). Under baseline conditions, we observed a range of $DF_{GABAA}$ polarities and magnitudes across different neurons, including hyperpolarizing, purely shunting, and depolarising $DF_{GABAA}$ (Fig. 6b, c). In a subset of mice, we performed continuous $DF_{GABAA}$ measurements before and after infusing 4-AP ($500\,\mu M$) into the cortex to elicit SLEs ("Methods"). The occurrence of SLEs was confirmed from the LFP recordings, and $DF_{GABAA}$ could be tracked in the same individual neurons before and after the SLE (Fig. 6d). This revealed that L1 GAD2+ interneurons exhibit depolarising shifts in $DF_{GABAA}$ as a result of SLEs (Fig. 6e). These data demonstrate that in vivo ORCHID affords easy and rapid reporting of resting and activity-dependent dynamic shifts in $DF_{GABAA}$ in the intact brain, under control and disease-relevant conditions.

## Discussion

All-optical approaches that combine voltage imaging and optogenetic stimulation represent a rapidly emerging strategy for studying neuronal dynamics in the intact nervous system[40,41]. Here we present the first all-optical strategy for studying ionic driving forces. We describe ORCHID - a powerful all-optical tool for the estimation of $DF_{GABAA}$, which represents the electromotive force for ion flux across inhibitory receptors. Whilst ORCHID could be used to estimate $DF_{GABAA}$ in any cell type or organ, we validate its use in the rodent nervous system. Taking advantage of ORCHID's ability to provide high-throughput, genetically targeted estimates of $DF_{GABAA}$ over extended periods of time, we confirm theoretical predictions regarding the mechanisms that establish $DF_{GABAA}$, utilise ORCHID to reveal cell-type-specific differences in activity-dependent $DF_{GABAA}$ changes during network activity, and provide the first in vivo measurements of intact $DF_{GABAA}$ in mouse cortical neurons.

ORCHID is easy to use with no spectral crosstalk between the orange-light excitation of Voltron2$_{608}$-ST, and blue-light activation of the integrated anion-permeable channel, GtACR2. This means that ORCHID is compatible with a wide variety of experimental paradigms. ORCHID's accuracy in reporting $DF_{GABAA}$ depends upon the size of the light-activated anion conductance (the GtACR2-induced change in the membrane conductance) relative to other conductances that are concurrently active in the cell of interest. For this reason, it is important that GtACR2 is known to generate large, consistent transmembrane conductances with minimal effects on neuronal function[27,42], and any future optimisations of light-activated anion channels are predicted to further improve ORCHID's accuracy. GtACR2's permeability sequence to different anions ($NO_3^- > I^- > Br^- > Cl^- > F^-$) is the same as other anion channels, including $GABA_A$Rs and GlyRs[27,43,44], which are also permeable to $HCO_3^-$, and exhibit a $Cl^-$-to-$HCO_3^-$ permeability ratio of five to one[43–46]. The equivalence of the anion permeabilities of GtACR2 and $GABA_A$Rs is supported by the fact that their reversal potentials are the same[47,48], and also our evidence that ORCHID measurements of $DF_{GABAA}$ are indistinguishable from those made by activating $GABA_A$Rs. ORCHID, therefore, provides an effective readout of the driving force across the receptors that mediate fast synaptic inhibition in the brain. As a result, it is possible that changes in intracellular pH and $HCO_3^-$, resulting in $DF_{HCO3-}$ differences between cell types or network states, could also contribute to observed differences in $DF_{GABAA}$ made using ORCHID. Nonetheless, as GtACR2 is primarily permeable to $Cl^-$[27], ORCHID measurements can also be considered to be estimates of $Cl^-$ driving force ($DF_{Cl}$), and future efforts to modulate the ion selectivity of light-activated $Cl^-$ channels could further enhance the accuracy of all-optical reporters of $DF_{Cl}$.

ORCHID's all-optical approach has several powerful advantages over currently available techniques for estimating $DF_{GABAA}$. Firstly, and most fundamentally, ORCHID does not perturb ionic gradients prior to measurement. Secondly, it is both genetically targetable and addressable by light, which affords excellent temporal and spatial control. Thirdly, ORCHID's high-throughput nature allows for measurements that are orders of magnitude faster than traditional techniques. A typical gramicidin-perforated patch-clamp recording requires 30–60 min per cell, compared to 1–5 s for an ORCHID recording. Fourthly, ORCHID affords measurements from small cells and potentially from subcellular compartments, which are difficult to access using electrode-based techniques. Fifthly, a strength of ORCHID is that the stability of recordings over time allows for the possibility of

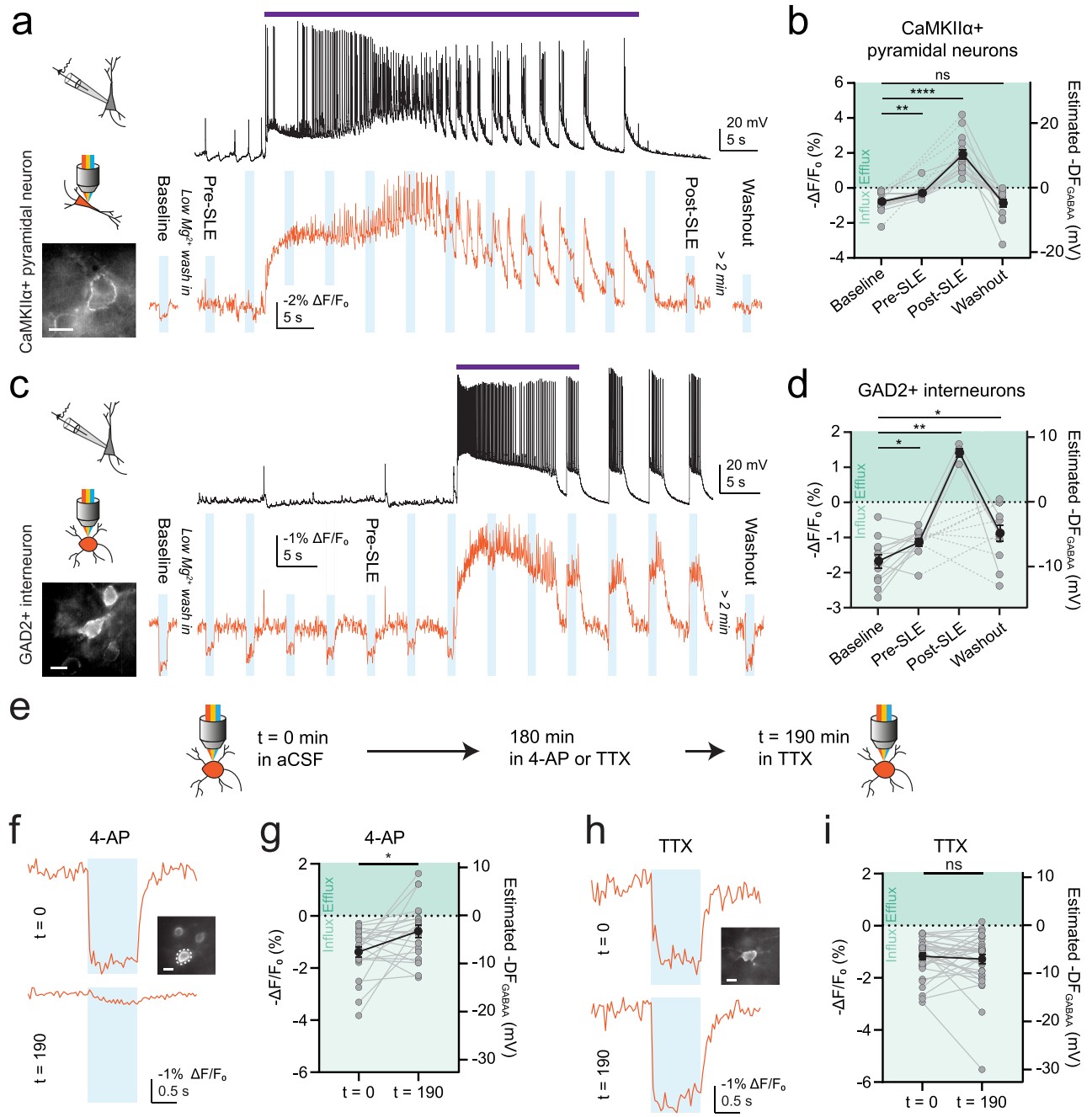

measuring relative changes in $DF_{GABAA}$, even if no switch in the polarity of the $DF_{GABAA}$ occurs.

In theory, a fluorescent reporter of $[Cl^-]_i$ could be used to report relative changes in a cell's $DF_{GABAA}$, by measuring changes in the amplitude of the $[Cl^-]_i$ response to an evoked $Cl^-$ conductance[49]. However, this is unlikely to be achieved in practice because of the inherent limitations in the sensitivity of $Cl^-$ imaging[25]. Furthermore, as fluorescent reporters of $[Cl^-]_i$ are cytoplasmic, detecting $[Cl^-]_i$ changes would require much larger and longer conductances that alter $[Cl^-]_i$ in the cytoplasm, unlike ORCHID whose readout only requires $Cl^-$ flux across the membrane to shift $V_m$. Finally, a fluorescent reporter of $[Cl^-]_i$ would not account for the $HCO_3^-$ component of $DF_{GABAA}$. By integrating a transmembrane voltage-reporter element and an anion conductance, ORCHID is highly sensitive to transmembrane flux and able to simultaneously read out $V_m$, both of which are key to making accurate measurements of $DF_{GABAA}$. Nonetheless, ORCHID is limited to

reporting $DF_{GABAA}$, and unlike other forms of $Cl^-$ imaging, cannot estimate absolute $[Cl^-]_i$ or $E_{GABAA}$. Further, it does not provide a measure of absolute $V_m$, although ratiometric GEVIs are being developed and could be integrated with future iterations of ORCHID to potentially report $DF_{GABAA}$ together with $V_m$ and, hence, $E_{GABAA}$[50].

Our results demonstrate how ORCHID strategies can be harnessed to test theoretical predictions about the underlying mechanisms that establish ionic driving forces. Our computational modelling and empirical evidence support the view that $DF_{GABAA}$ is primarily established by the action of CCCs such as KCC2[1]. Therefore, whilst genetically encoded $Cl^-$ indicators and other techniques that directly report $[Cl^-]_i$ are of considerable value, ORCHID highlights the danger of interpreting $[Cl^-]_i$ changes in the context of inhibitory synaptic transmission and $DF_{GABAA}$, without a concurrent readout of $V_m$. This is particularly the case for manipulations such as altering impermeant anions, which are predicted to shift $E_{Cl}$ or $E_{GABAA}$ and $V_m$ in tandem[32,51].

**Fig. 4 | ORCHID reveals dynamic inhibitory receptor driving forces over different timescales. a** ORCHID used with 1 s blue light stimulations to investigate activity-dependent variation in $DF_{GABAA}$ (bottom, orange trace) in a CaMKIIα+ pyramidal neuron (inset; scale bar: 10 μm) during low-$Mg^{2+}$ induced SLEs in hippocampal organotypic slices. Whole-cell current-clamp recording from a different CA1/CA3 pyramidal neuron provided an independent readout of SLEs (top, black trace; purple bar indicates SLE). The labelled blue-light stimulations indicate when the analysed measurements were taken. **b** Population data from CaMKIIα+ pyramidal neurons showing a significant shift in $DF_{GABAA}$ from baseline (before low-$Mg^{2+}$ aCSF wash-in) as compared to immediately pre-SLE (<15 s before SLE onset; Wilcoxon matched-pairs signed rank test, two-tailed, $P = 0.0020$, $n = 10$ cells) and immediately post-SLE (<15 s after SLE cessation; Wilcoxon matched-pairs signed rank test, two-tailed, $P = 0.000061$, $n = 15$ cells). However, >2 min post-SLE, $DF_{GABAA}$ had returned to baseline levels (baseline vs washout: Wilcoxon matched-pairs signed rank test, two-tailed, $P = 0.5995$, $n = 15$ cells). The dotted lines indicate cells for which pre-SLE $DF_{GABAA}$ was not recorded ($n = 5/15$ cells). **c** As in (**a**) but with GAD2+ interneurons targeted for ORCHID recording. **d** Population data of $DF_{GABAA}$ in GAD2+ interneurons showed a significant depolarising shift in $DF_{GABAA}$ between the baseline recordings and the recordings made pre-SLE (paired $t$ test, two-tailed, $P = 0.0136$, $n = 12$ cells). In 8/12 neurons, stimulation of GtACR2 as part of ORCHID caused further network burst events after the occurrence of an SLE, which persisted

for minutes (Supplementary Fig. 10). In the 4/12 recordings where GtACR2-induced network burst firing did not occur, there was a significant depolarising shift in $DF_{GABAA}$ post-SLE (paired $t$ test, two-tailed, $P = 0.0047$, $n = 4$ cells); dotted lines indicate cells in which network bursts occurred and post-SLE $DF_{GABAA}$ could not be recorded ($n = 8/12$ cells). When recorded again >2 min after the SLE, $DF_{GABAA}$ had returned to near baseline values (baseline vs washout, paired $t$ test, two-tailed, $P = 0.0317$, $n = 12$ cells). **e** Schematic of experimental design with ORCHID used to make paired recordings of $DF_{GABAA}$ 190 min apart in the same GAD2+ interneurons in hippocampal organotypic brain slices. **f** Top, baseline recording before a slice was incubated in 4-AP (50 μM) for 180 min. Bottom, recording after the slice had been transferred to aCSF containing TTX (1 μm) for 10 min. Inset: widefield fluorescence image of the recorded neuron; scale bar: 10 μm. **g** Population data showing $DF_{GABAA}$ depolarised significantly after a 180 min incubation in 4-AP (Wilcoxon matched-pairs signed rank test, two-tailed, $P = 0.0229$, $n = 22$ cells). **h** As in (**f**) but with recordings of $DF_{GABAA}$ before and after a 190 min incubation in TTX. Inset: widefield fluorescence image of the recorded neuron; scale bar: 10 μm. **i** Population data showing the magnitude of $DF_{GABAA}$ was not significantly different after a 190 min incubation in TTX (Wilcoxon matched-pairs signed rank test, two-tailed, $P = 0.5888$, $n = 34$ cells). ns = not significant ($P > 0.05$); *$P \leq 0.05$; error bars indicate mean ± SEM.

Given its ability to provide a sensitive readout of $DF_{GABAA}$, we note that ORCHID could serve as a high-throughput means for discovering new and improved enhancers, or inhibitors, of transmembrane $Cl^-$ transport.

By avoiding intracellular recordings, ORCHID enabled us to monitor unperturbed resting and dynamic $DF_{GABAA}$. We confirm predictions from previous work by showing that intense network activity can induce a profound change in $DF_{GABAA}$ polarity in neurons, with GABAergic synaptic input switching from inhibitory to excitatory, which is relevant for the management of status epilepticus[52,53]. Whilst SLE-associated $DF_{GABAA}$ estimates have been made using gramicidin-perforation in pyramidal neurons[17], we provide the first data showing that inhibitory interneurons also undergo an inversion in $DF_{GABAA}$ following seizure-like activity, which is predicted from observations that $Cl^-$ increases in interneurons during SLEs[54]. Recent work has suggested that the transition to seizures is associated with incremental increases in neuronal $[Cl^-]_i$, which is predicted to reduce $DF_{GABAA}$ prior to seizure onset[33]. We provide direct evidence for this clinically relevant phenomenon and thereby demonstrate the utility of ORCHID in dynamically estimating $DF_{GABAA}$ across network states.

By harnessing ORCHID's ability to be genetically targeted, we recorded $DF_{GABAA}$ from multiple cell types, including large populations of hippocampal CaMKIIα+ pyramidal neurons, GAD2+ interneurons, and GFAP+ astrocytes. This revealed that hippocampal astrocytes display a strong depolarising $DF_{GABAA}$ at rest, which is opposite to the hyperpolarizing $DF_{GABAA}$ exhibited by hippocampal pyramidal neurons in vitro, and is supported by the differential expression of CCCs between these two cell populations[17,36,48]. We observed that in contrast to neurons, astrocytic $V_m$ takes longer to respond to optogenetic activation, as has previously been described[55]; this should be considered when using ORCHID to estimate $DF_{GABAA}$ in this cell type. Previously, it has been suggested that astrocytes could serve as a source of extracellular anions that help to sustain the effectiveness of inhibitory GABAergic transmission[56], which has recently been supported by direct measurements of astrocytic $[Cl^-]_i$ during periods of network activity[16]. In support of this idea, our measurements of dynamic $DF_{GABAA}$ reveal that astrocytes are able to maintain a force for the outward flux of anions even during intense network activity. As other cell types exhibit specific expression patterns of CCCs and other anion transporters[57], ORCHID could be used to determine how these translate into differences in the anion-driving forces in cells such as oligodendrocytes, ependymal cells, microglia, and endothelial cells. Finally, we also demonstrate ORCHID's ability to provide

measurements of undisturbed $DF_{GABAA}$ in the intact brain. Our in vivo results captured a range of resting $DF_{GABAA}$ values in cortical L1 GAD2+ interneurons, which is different to what was observed in vitro. We hypothesise that this heterogeneity may be due, in part, to differences in network state[58] or the animal's sleep-wake history[15]. We revealed that $DF_{GABAA}$ is a dynamic parameter in the intact brain, with periods of intense activity causing substantial shifts towards depolarising $DF_{GABAA}$. Indeed, through the use of fibre photometry, or the implementation of a GEVI that is suitable for use with 2-photon microscopy, ORCHID has the potential to generate measurements of resting and dynamic $DF_{GABAA}$ throughout the mammalian brain[41], including from deep brain structures.

In terms of future applications, a further important question is how $DF_{GABAA}$ is regulated at a subcellular level. For example, it has been suggested that the properties of inhibitory synaptic input vary across a neuron and between different cellular compartments[17,59–63]. However, this has been difficult to address given the issue of space-clamp and additional technical challenges associated with electrode-based recordings. One could imagine combining ORCHID strategies with protein-targeting motifs and the patterned delivery of light in order to record $DF_{GABAA}$ from difficult-to-access subcellular compartments, such as distal apical dendrites, the axon initial segment, and astrocytic processes. In a similar manner, one could adopt targeted ORCHID strategies to estimate $DF_{GABAA}$/$DF_{Cl}$ across the membranes of organelles, such as mitochondria, lysosomes, and vesicles. In doing so, ORCHID could be used to study a range of cellular phenomena, including cell division, growth, and migration, in which anion/$Cl^-$ fluxes across different membranes contribute to the underlying intracellular and intercellular signalling processes.

In conclusion, we establish ORCHID as an all-optical strategy for estimating $DF_{GABAA}$ in the nervous system. This underscores the potential for similar all-optical strategies for elucidating how ionic driving forces affect cellular signalling in a range of biological contexts.

## Methods

### Ethics statement
The use of the animals in this study was approved by the University of Cape Town Animal Ethics Committee (AEC Protocol 021/026 and AEC Protocol 022/038).

### ORCHID construct subcloning
The ORCHID construct combined GtACR2 from the pAAV-EF1α-FRT-FLEX-GtACR2-EYFP plasmid, which was a gift from Mingshan Xue

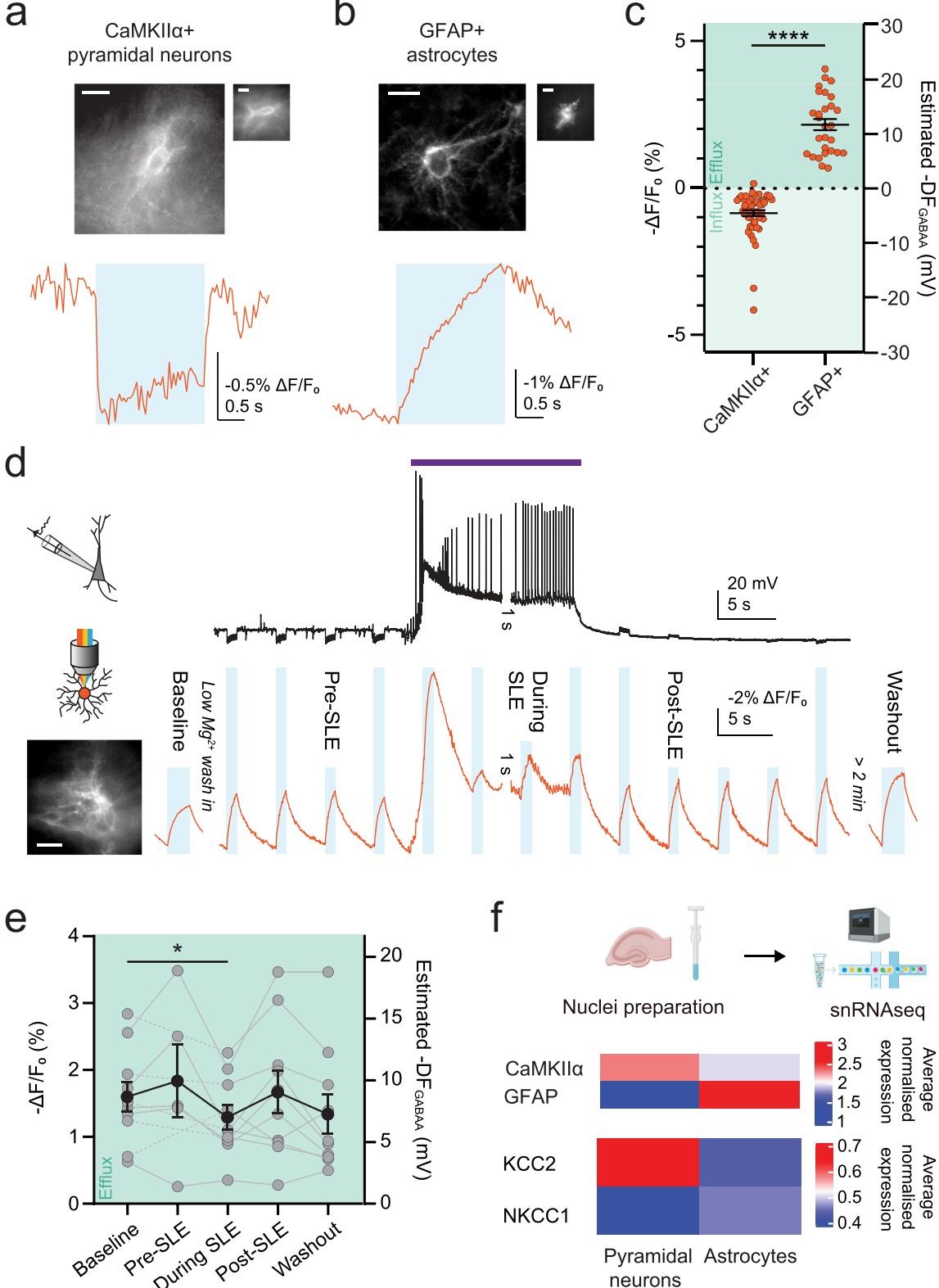

(Addgene plasmid #114369)[47], with Voltron2-ST from the pGP-pcDNA3.1 Puro-CAG-Voltron2-ST plasmid[26], which was kindly provided by Eric Schreiter and Ahmed Abdelfattah. The two genes were linked by a P2A linker sequence identical to that used by Holst et al.[64] to form an insert sequence, and include the Kozak sequence 5′ GCCACC(ATG) 3′. The vector used for the ORCHID construct originated from pAAV-hSyn-DIO {ChETA-mRuby2}on-W3SL, which was a gift from

Adam Kepecs (Addgene plasmid #111389)[65], with the hSyn promoter replaced with the EF-1α promoter from pAAV-nEF Con/Foff hChR2(H134R)-EYFP, which was a gift from Karl Deisseroth (Addgene plasmid #55647)[66]. The GtACR2-P2A-Voltron2-ST sequence of the ORCHID construct was synthesised in antisense (Thermo Fisher Scientific GeneArt Gene Synthesis) and inserted into the vector using the Bsp1407I/NheI restriction enzyme sites and the FastDigest versions of

**Fig. 5 | Astrocytes sustain an outward anion driving force during enhanced network activity.** ORCHID was used to compare $DF_{GABAA}$ in CaMKIIα+ pyramidal neurons (**a**) and GFAP+ astrocytes (**b**) in hippocampal organotypic slices. Widefield and confocal images of cells expressing ORCHID (top), and fluorescence recordings used to estimate $DF_{GABAA}$ from each cell type (bottom), with the cell that was recorded from shown inset if not in the main image; scale bar: 10 μm. **c** Estimated $DF_{GABAA}$ was significantly different between CaMKIIα+ pyramidal neurons and GFAP+ astrocytes ($4.52 \pm 0.57$ mV vs $-11.73 \pm 1.03$ mV; Mann-Whitney test, two-tailed, $P < 1 \times 10^{-15}$, $n = 51$ and 28 cells respectively). **d** ORCHID was used to investigate network-activity-dependent variation in $DF_{GABAA}$ (bottom, orange trace) in GFAP+ astrocytes (inset; scale bar: 10 μm) during low-$Mg^{2+}$ induced SLEs. The labelled blue-light stimulations indicate when the analysed measurements were taken. A whole-cell current-clamp recording from a CA1/CA3 pyramidal neuron provided a readout of network activity (top, black trace; purple bar indicates SLE). Note that there was a 1 s gap in the recording between the end of one recording, and the beginning of the next. **e** Population data of $DF_{GABAA}$ in GFAP+ astrocytes

showed depolarising $DF_{GABAA}$ measurements at baseline (before low-$Mg^{2+}$ wash-in) that were maintained pre-SLE (< 15 s before SLE onset; paired $t$ test, two-tailed, $P = 0.3098$, $n = 5$ cells), post-SLE (< 15 s after SLE cessation; paired $t$ test, two-tailed, $P = 0.5813$, $n = 10$ cells), and after washout (> 2 min after SLE cessation; Wilcoxon matched-pairs signed rank test, two-tailed, $P = 0.0840$, $n = 10$ cells). However, there was a small decrease in the magnitude of $DF_{GABAA}$ during the SLE (< 15 s before SLE cessation; baseline vs during SLE: paired $t$ test, $P = 0.0337$, $n = 10$ cells). Dotted lines indicate cells for which pre-SLE $DF_{GABAA}$ was not recorded ($n = 5/10$ cells). **f** Top, schematic showing nuclear dissociation and snRNAseq from mouse hippocampal organotypic brain slices performed using the 10X Genomics platform. Middle heatmap showing the average normalised expression of CaMKIIα and GFAP RNA in the pyramidal neuron and astrocyte clusters, respectively. Bottom, heatmap showing the average normalised gene expression between pyramidal neurons and astrocytes, which was different for KCC2 (*Slc12a5*, $P = 0.0324$, $n = 4$ samples) and NKCC1 (*Slc12a2*, $P = 0.0087$, $n = 4$ samples). -$DF_{GABAA}$ values plotted. ns = not significant ($P > 0.05$); *$P \le 0.05$; ****$P \le 0.0001$; error bars indicate mean ± SEM.

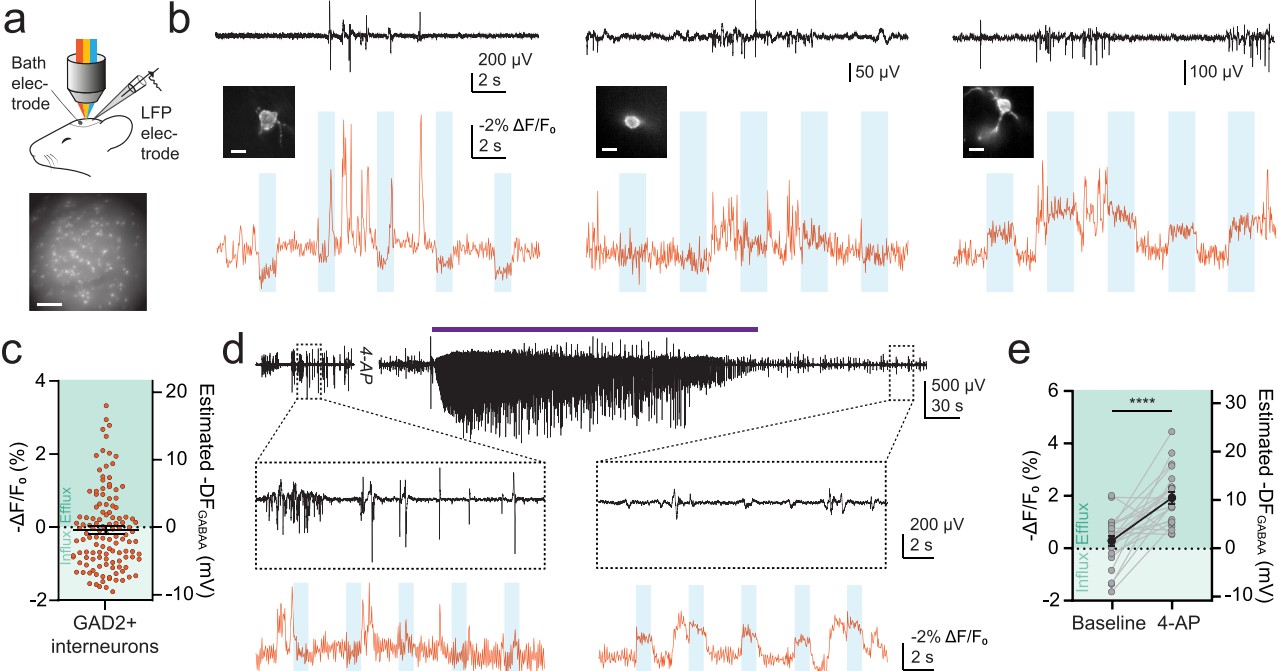

**Fig. 6 | In vivo determination of resting and dynamic inhibitory receptor driving forces in neurons. a** Schematic of the experimental setup (top) and a widefield fluorescence image of a region of L1 in S1 with GAD2+ interneurons expressing ORCHID (bottom), with similar images obtained in each mouse ($n = 10$ mice); scale bar: 100 μm. **b** LFP recordings were used to independently record population activity (top, black traces). ORCHID was used to concurrently record $DF_{GABAA}$ in GAD2+ interneurons in L1 of the anaesthetised mouse brain (bottom, orange traces), where a variety of different $DF_{GABAA}$ values were recorded, including hyperpolarizing (left), purely shunting (centre), and depolarising $DF_{GABAA}$ (right). Inset: widefield fluorescence images of the cells from which recordings were made;

scale bars: 10 μm. **c** Population data showing $DF_{GABAA}$ in L1 interneurons in S1 ($0.3873$ mV $\pm 0.5348$ mV, $n = 123$ cells from 10 mice). **d** 4-AP (500 μM) was infused into the cortex buffer in the recording chamber over the craniotomy to investigate $DF_{GABAA}$ dynamics during SLEs in vivo. ORCHID recording from a GAD2+ interneuron before (left) and after (right) a 4-AP induced SLE (bottom, orange traces). An LFP recording provided a readout of network activity (black traces, purple bar indicates SLE). **e** Population data of $DF_{GABAA}$ in GAD2+ interneurons showed a large depolarising shift in $DF_{GABAA}$ post-SLE (paired $t$ test, two-tailed $P = 0.000033$, $n = 22$ cells from 4 mice). -$DF_{GABAA}$ values plotted. ****$P \le 0.0001$; error bars indicate mean ± SEM.

the two enzymes (Thermo Fisher Scientific). Sanger sequencing confirmed the sequence of the final construct. Production of the AAV vector (serotype 1) containing the completed pAAV-EF1α-DIO-GtACR2-P2A-Voltron2ST-W3SL construct was carried out by Dr R. Jude Samulski and the University of North Carolina Vector Core. The ORCHID plasmid (pAAV-EF1α-DIO-GtACR2-P2A-Voltron2ST-W3SL) is available from Addgene (plasmid #217676).

### Animal husbandry
The animals used in this study were wild-type (C57BL/6 background) or GAD2-IRES-Cre mice (RRID: IMSR_JAX:010802), with the GAD2-IRES-

Cre strain being characterised by expression of Cre recombinase in all GABAergic interneurons. Male and female mice were group housed on a 12 h/12 h light/dark cycle, with the lights on at 06:00. Food and water were provided ad libitum. Mice of both sexes were used, although sex was not considered during the analysis.

### Mouse hippocampal organotypic brain slice cultures
Organotypic brain slice cultures were prepared using postnatal day (P) 7 mice and following the protocol originally described by Stoppini, Buchs and Muller[67] (for details, see Raimondo et al.[68]). All reagents were purchased from Sigma-Aldrich unless otherwise stated. Briefly,

the mice were killed by cervical dislocation, and the brains were removed and placed in 4 °C Earle's Balanced Salt Solution (EBSS), with 6.1 g/l HEPES, 6.6 g/l glucose, and 5% 1 M NaOH. The hippocampi were removed and sectioned into 350 μm slices using a tissue chopper (McIlwain). The slices were placed on Millicell-CM membranes and cultured in a medium consisting of (v/v): 25% EBSS; 49% minimum essential medium, 25% heat-inactivated horse serum; 1% B27 (Invitrogen, Life Technologies), and 6.2 g/l D-glucose. The slices were incubated at 35–37 °C in a 5% $CO_2$ humidified incubator.

Viral transduction was performed at 0–1 days in vitro (DIV). For experiments in which endogenous $GABA_ARs$ were activated, pGP-AAV-syn-flex-Voltron2-ST-WPRE-SV40 was utilised, while for ORCHID experiments, pAAV-EF1α-DIO-GtACR2-P2A-Voltron2ST-W3SL was utilised. AAV-CaMKIIα-Cre-GFP and AAV8-GFAP-GFP-Cre were used to target CaMKIIα+ pyramidal neurons and GFAP+ astrocytes, respectively. Additional targeting and validation were performed to ensure recordings were made from the correct cell types. Cells that were CaMKIIα+ were confirmed as being pyramidal neurons by their morphology (large apical dendrite extending into stratum radiatum) and the location of their cell bodies in the stratum pyramidale of CA1/CA3. Further, patch-clamp recordings provided electrophysiological confirmation in a subset of cells (Supplementary Fig. 13). GAD2+ cells were confirmed as being interneurons by their morphology (multiple large processes extending from the cell body) and the occurrence of their cell bodies in stratum radiatum. Further electrophysiological characterisation was used in a subset of cells (Supplementary Fig. 13). In addition, immunohistochemistry was used to confirm that DIO-ORCHID (i.e., DIO-GtACR2-P2A-Voltron2ST) and GAD1 were being co-expressed in the same cells in hippocampal organotypic slices from the GAD2-IRES-Cre mouse line (Supplementary Fig. 14; "Methods"). We confirmed the accurate targeting of astrocytes by GFAP-Cre through their morphological features (cells with small somata and multiple fine processes) and occurrence in the stratum radiatum. Additionally, patch-clamp recordings were performed from a subset of cells. In all cases ($n = 14/14$), we found a low resting $V_m$ ($-72.25 \pm 1.84$ mV), low membrane resistance ($59.76 \pm 12.53$ MΩ), and an inability to fire action potentials, which is typical of hippocampal astrocytes[69,70] (Supplementary Fig. 15). We also confirmed that repeated stimulation of astrocytes expressing GtACR2 did not affect baseline electrophysiological properties (Supplementary Fig. 15). In addition, at the frequency (0.2 Hz) and duration (1 s) at which $DF_{GABAA}$ was probed using ORCHID in GFAP+ astrocytes, we observed stable responses with no differences in the $DF_{GABAA}$ values measured within the same network state (either pre-SLE or post-SLE; Supplementary Table 3).

Here, as in all relevant methods, micropipettes were prepared from borosilicate glass capillaries (Warner Instruments). The AAVs were loaded into a glass micropipette along with FastGreen (0.1% w/v) to aid visualisation and were injected into the slices using the Openspritzer system[71].

Recordings were performed at 7–14 DIV, which is equivalent to P14-21 (as slices are prepared from P7 mice). This, as well as previous work, has shown that pyramidal neurons in the organotypic slices of the hippocampus have mature and stable $Cl^-$ homoeostasis mechanisms at this stage, as evidenced by their hyperpolarizing $DF_{GABAA}$[72–75]. Prior to imaging, slices were incubated for 30 min in aCSF bubbled with carbogen gas (95% $O_2$, 5% $CO_2$), containing either $JF_{549}$ or $JF_{608}$ HaloTag dye at 200 nM. The aCSF was composed of (in mM): NaCl (120), KCl (3), $MgCl_2$ (2), $CaCl_2$ (2), $NaH_2PO_4$ (1.2), $NaHCO_3$ (23), and D-glucose (11). During this incubation step, the dye binds irreversibly to the Voltron2-ST protein[28]. After the incubation was complete, slices were transferred to a submerged recording chamber where they were continuously superfused with 32 °C carbogen-bubbled aCSF using a peristaltic pump (Watson-Marlow). Pharmacological agents included CGP-55845 (5 μM) and 2-chloroadenosine (4 μM) in the recording aCSF for all experiments in which $GABA_ARs$ were activated, as well as

picrotoxin (PTX; 100 μM), VU0463271 (VU; 10 μM), 4-aminopyridine (4-AP; 50 μM), and tetrodotoxin (TTX; 1 μM) for subsets of the in vitro experiments.

## Surgical procedures for in vivo imaging

Intracranial bulk regional viral injections of pAAV-nEF-DIO-GtACR2-P2A-Voltron2-ST-W3SL were performed on P0-1 GAD2-IRES-Cre mouse pups[76,77]. Adult breeders were removed from the home cage and placed in a holding cage for the duration of the procedure. A 30 G insulin syringe was loaded with 1 μl AAV combined with FastGreen (0.1% w/v) to aid visualisation and mounted in a stereotaxic frame. Pups were individually removed from their home cage and anaesthetised using 2 ml isoflurane pipetted onto a gauze swab in an enclosed space. Respiration rate and skin colour were monitored to assess the depth of anaesthesia. Once the depth of anaesthesia was sufficient, pups were rapidly transferred to a heating pad beneath the stereotaxic frame and the AAV was injected into S1 of the left cerebral hemisphere at a depth of ~1 mm, with a successful injection indicated by the green coloration of the hemisphere. The injection typically took ~30 s. The pup was then transferred back to its home cage and monitored until fully recovered from anaesthesia when the injection of the next pup was begun. We observed a 100% survival rate of injected pups using this method.

Imaging experiments began after P28. Retroorbital injections of $JF_{608}$ were performed 3–24 h prior to imaging[26,28]. 100 nmol $JF_{608}$ was dissolved in 20 μl DMSO, 20 μl Pluronic F-127 (20% w/v in DMSO, Thermo Fisher Scientific), and 60 μl sterile phosphate-buffered saline (PBS). The mice were anaesthetised using an intraperitoneal (IP) injection of ketamine/xylazine mixture (ketamine 80 mg/kg and xylazine 10 mg/kg). The $JF_{608}$ solution was delivered using a 30 G insulin syringe as described by Yardeni et al.[78]. The preparation for anaesthetised recordings was adapted from Burman et al.[58]. Mice were anaesthetised using 3 l/s $O_2$ with 5% isoflurane for induction, and with the isoflurane reduced to 1–2% for maintenance. Once anaesthetised, the animal was secured in a stereotaxic frame (Narishige). An IP injection of buprenorphine (0.5 mg/kg) and a subcutaneous (SC) injection of 200 μl sterile saline were administered for analgesia and hydration, respectively. The mouse's body temperature was maintained at ~37 °C using a heating pad and a temperature probe. The head was shaved, and eye-protecting ointment (Duratears) was applied to each eye. An incision in the scalp was made using surgical scissors, and the area was expanded by blunt dissection to expose the skull. Forceps were used to remove any membranous tissue from the surface of the skull. Tissue adhesive (Vetbond) was used to fix the edges of the incised scalp to the skull and to stabilise the cranial sutures. Several layers of dental cement (InterDent) were applied to create a recording chamber on top of the skull. A 0.5 mm craniotomy was drilled over S1 using a dental drill (Dental Lab). The craniotomy was submerged in cortex buffer containing (in mM): NaCl (125), KCl (5), HEPES (10), $MgSO_4 \cdot 7H_2O$ (2), $CaCl_2 \cdot 2H_2O$ (2), and D-glucose (10). The bone flap and dura were removed. The mouse was then transferred to the imaging setup. In a subset of animals, 4-AP was infused into the cortex buffer in the recording chamber using a pipette (500 μM working concentration). The recording session typically lasted 3 hours.

## Electrophysiology

For patch-clamp recordings, micropipettes were filled with a low-$Cl^-$ internal solution composed of (in mM): K-gluconate (120), KCl (10), $Na_2ATP$ (4), NaGTP (0.3), Na2-phosphocreatine (10), and HEPES (10). For high-$Cl^-$ recordings, pipettes were filled with a high-$Cl^-$ internal solution ($[Cl^-] = 141$ mM) composed of (in mM): KCl (135), NaCl (8.9), and HEPES (10). For all gramicidin-perforated recordings, gramicidin A from *Bacillus brevis* was added to the internal solution for a working concentration of 80 μg/ml. Adequate perforation of gramicidin-perforated recordings was assessed by monitoring access resistance

and was defined as when access resistance was < 90 MΩ. Recordings were made using an Axopatch 200B amplifier (Molecular Devices), and data was acquired using WinWCP version 5.6.6 (University of Strathclyde). To calculate $R_m$ in current-clamp mode, the change to $V_m$ upon injection of 600 pA was used to calculate $R_{total}$, from which the pipette resistance was then subtracted. For experiments investigating the role of impermeant anions in setting $DF_{GABAA}$, anionic 10 000 MW dextran bound to Alexa-Flour 488 (Thermo Fisher Scientific) was electroporated into cells. This molecule is a hydrophilic polysaccharide, being both membrane impermeant and having a large negative average charge. A micropipette was filled with 5% dextran solution in PBS, and the micropipette was positioned near the soma of the targeted neuron. Voltage pulses (5–10 pulses, 20 ms duration, 0.5–1 V) were applied using a stimulus isolator, and successful electroporation was confirmed visually by observing the neuron filled with the fluorescent dye. For in vivo LFP recordings, micropipettes were filled with cortex buffer. The tip of the pipette was broken against tissue paper with the aim of achieving a tip diameter of ~ 5 μm. Recordings were made using a differential amplifier (A-M system, 1800), and data was acquired using LabChart 8 (ADInstruments).

## Widefield fluorescence imaging

Neurons were visualised using a BX51WI upright microscope (Olympus) equipped with a 20x water-immersion objective (Olympus XLUMPlanFL) and a sCMOS camera (Andor Zyla 4.2). For imaging Voltron2$_{549}$-ST, a green LED with a 555/28 nm excitation filter (Lumencor Aura III) was used for excitation (20.6 mW/mm²) with a further multi-band filter set for collecting emission between 590 and 660 nm (Chroma, 69401-ET-380/55-470/30-557/35 Multi LED set). For Voltron2$_{608}$-ST or ORCHID, a high-powered orange LED (590 nm, SOLIS-590C, Thorlabs) was used for excitation (143 mW/mm²). Optogenetic activation of GtACR2 as a part of ORCHID was achieved using a blue LED (475/28 nm, Lumencor Aura III), with the intensity of this LED fixed at 3.6 mW/mm². A T570lpxr beam splitter was used for combining the blue and orange LEDs before a filter set with a 599/13 nm excitation filter, a 612 nm long pass beam splitter, and a 632/28 nm emission filter was used (Chroma, 49311-ET-Red#3 Narrow band FISH).

Illumination of the tissue was restricted to a ~ 140 μm diameter region using the built-in Olympus variable-diameter field stop aperture. Image acquisition was controlled using μManager version 2.0.0. For current-step recordings, exposure was 5 ms with an image acquisition frequency of 200 Hz. For all voltage-step recordings and Voltron2$_{549}$-ST recordings, exposure was 10 ms and image acquisition frequency was 100 Hz. For ORCHID recordings, exposure was 20 ms and image acquisition frequency was 25 Hz. The imaging region was 256 × 256 pixels, and with no overlap between successive camera exposures, this ensured a 20 ms gap between successive exposures. Transistor-transistor logic (TTL) signals from the camera were set to output throughout the exposure, and this signal was sent to an Arduino that ran a custom script to activate the blue LED immediately after each camera exposure ended and for 15 ms, allowing for a 5 ms gap between the blue LED being switched off and the subsequent camera exposure. We found this to be crucial for artefact-free imaging. Camera acquisition and blue-light activation were thus both at a frequency of 25 Hz, with 50% and 37.5% on time, respectively.

To calculate the duration of optogenetic stimulation during ORCHID recordings with a frame rate of 25 Hz, we aimed to detect a typical $\Delta F/F_0$ of 1% (Eq. 1), with an SNR of ~ 10 per optogenetic stimulation. We assumed negligible dark noise and read noise and accounted for Poisson-distributed shot noise with a value of $\sqrt{F_0}$ (Eq. 2). The Andor Zyla 4.2 sCMOS camera that was used has a well depth (e⁻) of 30000, and a maximum quantum efficiency of 82%, giving a required number of e⁻ per frame of 36585.37. Thus, $F_O$ can be calculated as

follows:

$$\frac{\Delta F}{F_0} = \frac{1}{100} \tag{1}$$

$$SNR = \frac{\Delta F}{\sqrt{F_0}} \tag{2}$$

$$F_0 = 1 \times 10^6 \tag{3}$$

Therefore, the number of frames required per optogenetic stimulation to achieve an SNR of 10 at a $\Delta F/F_0$ of 1% is:

$$\frac{1 \times 10^6}{36585.37} = \sim 27 \tag{4}$$

We used an optogenetic stimulation time of 1 s for longer recordings (which gives an expected SNR of ~ 9.6 for a $\Delta F/F_0$ of 1% at 25 Hz frame rate), and for shorter recordings, we used a stimulation time of 2 s (which gives an expected SNR of ~ 13.5 for a $\Delta F/F_0$ of 1% at 25 Hz frame rate). These times were additionally chosen to have a similar total time of optogenetic activation in the shorter recordings and the longer recordings (10 s vs 13 s).

When utilising Voltron2$_{549}$-ST with GABA$_A$R activation, 500 μM GABA in recording aCSF was loaded into a micropipette, and 100 ms puffs of synthetic air were delivered to the micropipette using the Openspritzer system. This duration was chosen to maximise GABA delivery while minimising tissue movement. Imaging traces were 7 s long with a 1 s baseline period prior to the puff. Care was taken to ensure no movement artefact from the puff. ORCHID recordings had a duration of 25 s and consisted of five 1 s or 2 s strobed optogenetic activations, which were averaged post-hoc for some experiments (see Data Analysis). These durations are broadly similar to the duration of GABA$_A$R stimulation during micropipette delivery of GABA. During low-Mg²⁺ wash-ins, in order to increase the chance of recording an SLE, the duration of recordings was increased to 65 s, and ORCHID was activated with 1 s periods of strobed light every 5 s. Here the averaging of multiple ORCHID activations was not possible due to the rapid $DF_{GABAA}$ changes. Instead, we took the mean signal during a single optogenetic activation (1 s) during each relevant period, which was used alongside a baseline period immediately prior to the optogenetic activation. The relevant periods were: < 15 s before SLE onset for pre-SLE, < 15 s before SLE cessation for during SLE, and < 15 s after SLE cessation for post-SLE. For pre-SLE and post-SLE measurements, optogenetic activations where neither the activation nor the baseline period was contaminated by SLE-associated spiking/depolarisations were selected (which excluded any measurements during GtACR2-triggered network burst firing). For during-SLE measurements in astrocytes, a measurement close to the end of the SLE, where measurements were typically more consistent and thus representative, was selected. When testing the stability of $DF_{GABAA}$ measurements made using ORCHID in astrocytes within the same network state (either pre-SLE or post-SLE), the same paradigm of ORCHID measurements made using 1 s optogenetic activations every 5 s was used, with a statistical comparison between measurements of $DF_{GABAA}$ made 20 s apart. SLEs were defined as events with significant deviation from the resting potential in patch-clamp recordings (> 2 standard deviations) lasting for at least 5 s. Network burst events that occurred post-SLE upon GtACR2 stimulation were defined as constant depolarisations of > 10 mV above baseline that lasted for longer than 250 ms.

## Single-nucleus RNA sequencing

Nuclei from mouse hippocampal organotypic slices were dissociated in lysis buffer (Nuclei EZ Prep, NUC101) on ice and processed via the

10X Genomics platform to prepare libraries for sequencing as outlined in the Chromium Next GEM 3' Single Cell 3 Reagent Kits v3.1 user guide. Four samples, each comprising 36 hippocampal slices, were processed. Libraries were sequenced using an Illumina NovaSeq 6000 S2 flow cell. Analysis of the snRNAseq data was performed on facilities provided by the University of Cape Town's ICTS High-Performance Computing team. Cell Ranger version 7.1.0 was used to map paired-end sequencing reads to the mouse reference transcriptome (refdata-gex-mm10-2020-A). The filtered feature-barcode matrix of each sample was converted into a Seurat object. Samples were processed according to the standard Seurat pipeline in R version 4.0.5 or 4.2.0[79].

Quality control (QC) involved removing nuclei with less than 500 unique molecular identifiers (UMIs), fewer than 250 genes expressed, log10GenesPerUMI less than 0.8, and/or a mitochondrial ratio greater than 0.2. In addition, all mitochondrial genes were excluded from the dataset. Three doublet identification tools were employed, including DoubletFinder[80], Scrublet[81], and DoubletDecon[82]. For each sample, all doublets called by DoubletFinder, together with the intersection of doublets called by Scrublet and DoubletDecon, were filtered out. Following QC, normalisation and variance stabilisation were performed using the SCTransform() function. The mitochondrial ratio was regressed during this step. Integration of the different samples was performed via Seurat's FindIntegrationAnchors() function using the top 2000 most variable features and 30 dimensions followed by the IntegrateData() function. A principle component analysis (PCA) was run on the integrated data followed by clustering using the FindNeighbors() and FindClusters() functions.

Several cluster annotation approaches were employed, including automated annotation, manual annotation, and Seurat's label transfer method. For automated annotation, Seurat's FindAllMarkers() function was used to identify differentially expressed genes (DEGs) between the clusters. A cluster resolution of 0.4 was set as the active identity and comprised 30 clusters to be annotated. An automated annotation tool known as SCSA[83] was used to obtain putative cluster annotations by comparing the list of DEGs per cluster obtained from FindAllMarkers() to reference datasets of known cell-type-specific markers, including SCSA's built-in reference databases[84], as well as user-defined databases curated from mousebrain.org[85]. Manual inspection methods involved visualising the expression of a smaller set of known cell-type-specific markers across the different clusters in bubble plots, as well as searching for the cell-type-specific markers in the list of cluster-specific DEGs from the FindAllMarkers() output. Using the automated and manual annotation methods, putative user-defined annotations for each cluster were generated. In addition, Seurat's label transfer method was carried out using the Allen Mouse Brain atlas as the reference[57]. Seurat's FindTransferAnchors() function was used to find anchors between the reference and query datasets, with SCT specified as the normalisation method and the top 30 dimensions used. A relatively stringent filtering step was then performed to remove any nuclei that did not have an agreement between their broad user-defined and Allen Mouse Brain annotations. Following this filtering step, the user-defined coarse annotations were used for downstream analyses. The 4 samples were subsetted, and heatmaps were plotted to visualise the normalised average expression of several genes of interest across the pyramidal neuron and astrocyte clusters, respectively.

## Immunohistochemistry

Mouse hippocampal organotypic brain slices (DIV 12) were incubated in 1 mM $JF_{608}$ dye in 2 ml carbogen-bubbled aCSF for 30 min and then transferred to a recording chamber perfused with aCSF for 1 h to allow the unbound dye to wash out. Slices were fixed in 4% paraformaldehyde in PBS for 15 min at 4 °C and permeabilized using wash buffer (0.3% Triton X-100 in PBS) for 30 min at 4 °C. Next, the slices were incubated in blocking solution (3% bovine serum albumin and 1% Triton X-100 in PBS) at room temperature (RT) for 4 h. This was followed by overnight incubation at 4 °C with rabbit anti-mouse GAD1 polyclonal antibody (Proteintech, catalogue number: 10408-1-AP) diluted in blocking solution (1:400). Slices were then washed three times and incubated with donkey anti-rabbit IgG conjugated to Alexa Fluor 488 (Abcam, catalogue number: ab150073; 1:500) in blocking solution for 4 h at RT. This was followed by three wash steps and nuclear staining with Hoechst (1:5000) in blocking solution for 15 min at RT. The slices were washed again and mounted.

## Confocal imaging

Confocal images of live and fixed brain slices were obtained on an LSM 880 Airyscan confocal microscope (Carl Zeiss, ZEN SP 2 software) using a C-Apochromat 40 × objective. To visualise Hoechst a 405 nm laser was used, to visualise Alexa Fluor 488 a 488 nm laser was used, and to visualise ORCHID-expressing cells a 561 nm laser was used.

## Data analysis

Data analysis was performed using custom scripts written in MATLAB R2022b (MathWorks). The image analysis pipeline extracted signals from a region of interest (ROI) targeted to the soma of the cell and calculated an integrated density of fluorescence (i.e., mean pixel brightness) in this region ($F_s$). Background subtraction was performed by selecting an ROI targeted to a representative area of the background, calculating an integrated density of fluorescence in this region, and subtracting this from $F_s$ image-wise. For ORCHID recordings, this background fluorescence signal was smoothed with a smoothing window of 100 before being subtracted from $F_s$. Correction for photobleaching and low-frequency changes in signal was performed by fitting and subtracting a 9th- or 10th-order polynomial to and from the background-corrected $F_s$ trace. The response to the GABA puff or optogenetic stimulation in the $F_s$ trace was excluded from the fitting of this polynomial to avoid removing the signal from the trace. For recordings using Voltron2$_{549}$-ST with GABA$_A$R activation, this corrected $F_s$ trace was smoothed with a smoothing window of 7, and $\Delta F/F_0$ values were calculated using the maximum or minimum value of the smoothed signal post-puff and the mean of the smoothed baseline period prior to the puff. This same process was used for ORCHID recordings made during activity-dependent variation in DF$_{GABAA}$ experiments, with the maximum or minimum value of the smoothed signal during an optogenetic stimulation being used to calculate $\Delta F/F_0$. For all other in vitro ORCHID recordings, the 5 strobed optogenetic stimulation periods in the $F_s$ trace were averaged before $\Delta F/F_0$ was calculated using the mean during this averaged optogenetic stimulation, and the mean during the averaged baseline period prior to optogenetic stimulation onset. Due to the higher levels of spontaneous activity in vivo, for in vivo ORCHID recordings $\Delta F$ and $F_0$ were calculated using the mean fluorescence during a single optogenetic stimulus and an equivalent duration of recording on one side of the stimulus (these values termed $\Delta F_S$ and $F_{0S}$). To correct for neuropil contamination, the indices of these two periods (the period during the stimulus and the baseline period) were used to calculate an equivalent $\Delta F$ from the background fluorescence trace ($\Delta F_{BG}$), which was subtracted from $\Delta F_S$ (giving $\Delta F_{S-BG}$). $\Delta F/F_0$ was then calculated using $\Delta F_{S-BG}$ and $F_{0S}$. The relationship between $\Delta V$ and $\Delta F/F_0$ for Voltron2$_{608}$-ST and for Voltron2$_{549}$-ST, which were acquired and averaged from multiple simultaneously patched and imaged neurons, was linear (Fig. 1i, j and Supplementary Fig. 4c, d). Subsequently, a measured $\Delta F/F_0$ in cells imaged but not patched could be translated to a $\Delta V$, as previously performed by Cornejo et al.[30]. This translated $\Delta V$ was then used to estimate DF$_{GABAA}$. This analysis pipeline used the WinWCP MATLAB Importer written by David[86].

## Computational model

We simulated a neuron, using the pump-leak mechanism, as a single compartment within an extracellular environment consisting of fixed ion concentrations (i.e., an infinite bath model). The compartment was modelled as a cylinder with a diameter of 1 μm and a length of 25 μm. The ionic flow between the compartment and external environment was permitted via leak channels, Na$^+$/K$^+$-ATPase, and KCC2 transporters. For details of the simulation equations for ionic flux and volume change, see Düsterwald et al.[32]. In addition, GABA$_A$Rs, $HCO_3^-$, and pH regulation were added to the model in Düsterwald et al., which we describe below. All parameters are given in Table 1.

**Table 1 | Computational model parameters**

| Symbol | Value | Description |
|---|---|---|
| Constants | | |
| $F$ | 96485.33 C/mol | Faraday constant |
| $R$ | 8.31446 J/(K·mol) | Universal gas constant |
| $T$ | 310.15 K | Absolute temperature (37 °C) |
| Parameters | | |
| Radius | 5 μm | The radius of the compartment |
| Length | 25 μm | Length of the compartment |
| $C_m$ | $2 \times 10^{-6}$ F/cm$^2$ | Unit membrane capacitance |
| $g_{Na}$ | 25 μS/cm$^2$ | Na$^+$ leak conductance |
| $g_K$ | 70 μS/cm$^2$ | K$^+$ leak conductance |
| $g_{Cl}$ | 20 μS/cm$^2$ | Cl$^-$ leak conductance |
| $g_{HCO3}$ | 4 μS/cm$^2$ | $HCO_3^-$ leak conductance |
| $g_{KCC2}$ | 20 μS/cm$^2$ | KCC2 conductance |
| $v_w$ | 0.018 dm$^3$/mol | Partial molar volume of water[88] |
| $p_w$ | 0.0015 dm/s | Osmotic permeability[88] |
| $P$ | 0.1 C/(dm$^2$·s) | ATPase pump rate constant |
| PCO$_2$ | 0.05 atm | Partial pressure of CO$_2$ |
| $k_H$ | 0.031 M/atm | Henry's law constant for CO$_2$ in water |
| $K_f$ | $10^3$ s$^{-1}$ | Forward rate constant |
| $K_r$ | $2.539 \times 10^9$ s$^{-1}$.M$^{-1}$ | Reverse rate constant |
| $g_{GABAA\_max}$ | 10 nS | Maximum conductance of GABA$_A$Rs |
| pH$_o$ | 7.4 | Extracellular pH |
| pH$_i$ | 7.2 | Intracellular pH |
| $[K^+]_o$ | 3.5 mM | Extracellular K$^+$ concentration |
| $[Na^+]_o$ | 145 mM | Extracellular Na$^+$ concentration |
| $[Cl^-]_o$ | 110 mM | Extracellular Cl$^-$ concentration |
| $[HCO_3^-]_o$ | 31 mM | Extracellular $HCO_3^-$ concentration |
| $[H_2CO_3]_o$ | 1.55 mM | Extracellular $H_2CO_3$ concentration |
| $[H_2CO_3]_i$ | 1.55 mM | Intracellular $H_2CO_3$ concentration |
| $[X^-]_o$ | 7.5 mM | Extracellular impermeant anion concentration |
| $z_o$ | −1 | Extracellular impermeant anion average charge |
| $z_i$ | −0.85 | Intracellular impermeant anion average charge[89] |
| Variables (default steady state) | | |
| $[K^+]_i$ | 122.6 mM | Initial intracellular K$^+$ concentration |
| $[Na^+]_i$ | 15 mM | Initial intracellular Na$^+$ concentration |
| $[Cl^-]_i$ | 5.3 mM | Initial intracellular Cl$^-$ concentration |
| $[HCO_3^-]_i$ | 9.7 mM | Initial intracellular $HCO_3^-$ concentration |
| $[X^-]_i$ | 144.4 mM | Initial intracellular impermeant anion concentration |
| $V_m$ | −67.2 mV | Initial membrane potential |
| $w$ | 19 pL | Compartment volume |

Fast GABAergic synaptic transmission mediated by GABA$_A$Rs was modelled as an alpha function with a tau ($\tau$) of 250 ms, and a max conductance $g_{GABAA\_max}$ of 10 nS, for $t > t_{onset}$:

$$g_{GABAA} = g_{GABAA\_max} \cdot \frac{t - t_{onset}}{\tau} \cdot e^{\left[\frac{-(t - t_{onset} - \tau)}{\tau}\right]} \tag{5}$$

Current through GABA$_A$Rs ($I_{GABAA}$) was modelled as follows:

$$I_{GABAA} = I_{GABA_{Cl^-}} + I_{GABA_{HCO_3^-}} \tag{6}$$

$$I_{GABA_{Cl^-}} = \chi \cdot g_{GABAA} \cdot (V_m - E_{Cl^-}) \tag{7}$$

$$I_{GABA\_HCO_3^-} = (1 - \chi) \cdot g_{GABAA} \cdot \left(V_m - E_{HCO_3^-}\right) \tag{8}$$

$\chi$ represents the fraction of the total GABAergic current carried by Cl$^-$ and is given by:

$$\chi = \frac{E_{HCO_3^-} - E_{GABAA}}{E_{HCO_3^-} - E_{Cl^-}} \tag{9}$$

The reversal potentials $E_{Cl^-}$, $E_{HCO_3^-}$, and $E_{GABAA}$ were each updated throughout the simulation using the respective Nernst and Goldman-Hodgkin-Katz equations:

$$E_{Cl^-} = \frac{R \cdot T}{F} \ln \left( \frac{[Cl^-]_i}{[Cl^-]_o} \right) \tag{10}$$

$$E_{HCO_3^-} = \frac{R \cdot T}{F} \ln \left( \frac{[HCO_3^-]_i}{[HCO_3^-]_o} \right) \tag{11}$$

$$E_{GABAA} = \frac{R \cdot T}{F} \ln \left( \frac{\frac{4}{5}[Cl^-]_i + \frac{1}{5}[HCO_3^-]_i}{\frac{4}{5}[Cl^-]_o + \frac{1}{5}[HCO_3^-]_o} \right) \tag{12}$$

Here, $R$ is the ideal gas constant, $F$ is Faraday's constant, and $T$ is temperature.

CO$_2$ was assumed to be equilibrium-distributed across the simulated plasma membrane. For a partial pressure of CO$_2$ (PCO$_2$) of 38 mmHg (5% CO$_2$), [CO$_2$] was calculated using Henry's Law and a value for Henry's Law constant ($k_H$) for CO$_2$ in the water of 0.031 (M/atm):

$$[CO_2] = k_H \cdot PCO_2 \tag{13}$$

Assuming that [CO$_2$] ≈ [H$_2$CO$_3$], [H$_2$CO$_3$] (inside and outside the compartment) was calculated as 1.55 mM.

The CO$_2$ hydration reaction outside the compartment was assumed to be under equilibrium:

$$H_2O + CO_2 \rightleftharpoons H_2CO_3 \rightleftharpoons H^+ + HCO_3^- \tag{14}$$

Using the Henderson Hasselbalch equation and a pK$_a$ value of 6.1 as the dissociation constant for H$_2$CO$_3$, [HCO$_3^-$]$_o$ was calculated at a pH of 7.4:

$$[HCO_3^-]_o = [H_2CO_3] \cdot 10^{pH - pK_a} \tag{15}$$

This gives a [HCO$_3^-$]$_o$ of 31 mM.

Inside the compartment, pH was held constant at 7.2, and the production HCO$_3^-$ by the forward reaction of the CO$_2$ hydration reaction (Eq. 14) was calculated using the forward rate equation and a

forward rate constant $K_f$ of $1 \times 10^3$:

$$\text{Rate}_{\text{forward}} = K_f \cdot [\text{H}_2\text{CO}_3] \qquad (16)$$

The removal of $\text{HCO}_3^-$ by the reverse reaction was calculated using the reverse rate equation and a reverse rate constant $K_r$ of $2.539 \times 10^9$:

$$\text{Rate}_{\text{reverse}} = K_r \cdot [\text{H}^+] \cdot [\text{HCO}_3^-] \qquad (17)$$

$\text{H}^+$ produced or removed by the forward or reverse reactions was simulated as being immediately exchanged with $\text{Na}^+$ via the $\text{Na}^+/\text{H}^+$ exchanger[87] in order to maintain intracellular pH at 7.2.

## Statistics & reproducibility
Statistical measurements were performed using GraphPad Prism 7. Data were assessed for normality, and parametric or non-parametric tests were selected as required. Data is reported as mean ± SEM unless stated otherwise. No statistical method was used to predetermine the sample size. Sample sizes were chosen based on accepted standards in the field, aiming to minimise the number of animals sacrificed while still being able to demonstrate the robustness of any effects. No data were excluded from the analyses, with three predetermined exceptions. For ORCHID recordings, cells which were visually determined to have blebbed during recordings were excluded. For in vitro experimental data gathered using soma-directed puffs of GABA, recordings in which a movement artefact was present were excluded. Patch-clamp recordings in which the access resistance was greater than 30 MΩ were excluded. Mice were randomly selected for hippocampal slices or imaging experiments and were randomly assigned to experimental groups. The selection of cells for imaging was random, and within-cell comparisons were performed wherever possible. Investigators were blinded to experimental conditions where possible during data acquisition and analysis, although morphological differences between cells and technical differences between techniques meant that blinding was often not possible. Voltage imaging and electrophysiology experiments were performed independently for each individual cell in an experimental dataset.

## Reporting summary
Further information on research design is available in the Nature Portfolio Reporting Summary linked to this article.

## Data availability
Source data are provided with this paper as a Source Data file. Imaging raw data are available upon request, due to the size of the dataset. The expected timeframe for response to access requests is approximately two weeks, accounting for communication and upload time, and depending on the size of the data requested. Source data are provided in this paper.

## Code availability
All data was analysed using custom scripts created in MATLAB, which are available at: https://github.com/joshs08/ORCHID (https://doi.org/10.5281/zenodo.13684771). Code used for the simulations is accessible at: https://github.com/Eran707/Single-Cell-Simulator (https://doi.org/10.5281/zenodo.13376723).

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

## Acknowledgements
We would like to thank members of the Raimondo lab for advice and comments and Suzanne Duncan for assistance in naming ORCHID. We thank Sarah E. Plutkis, Katie L. Holland, Jonathan B. Grimm, and Luke D. Lavis (Janelia Research Campus) for Janelia Fluor HaloTag ligands, and Yoav Adam for providing advice on GEVI imaging. We thank Rodney Lucas for assistance in performing animal experiments, the Africa Microscopy Initiative Imaging Centre for their support and assistance with imaging, and Christina Steyn, Dorit Hockman, Muazzam Jacobs and Christopher Dulla for advice and support in performing snRNAseq analysis. The research leading to these results has received support from the Gabriel Foundation (J.V.R.), a Wellcome Trust Seed Award (214042/Z/18/Z) (J.V.R.), the FLAIR Fellowship Programme (FLR \R1\190829): a partnership between the African Academy of Sciences and the Royal Society funded by the UK Government's Global Challenges Research Fund (JVR), a Wellcome Trust International Intermediate Fellowship (222968/Z/21/Z) (J.V.R.), ERC Grant Agreement 617670 (C.J.A.), and the project was supported by funding from BBSRC project BB/S007938/1 (C.J.A.).

## Author contributions
J.S.S., A.S.A., E.R.S., C.J.A., and J.V.R. conceptualised the study. A.S.A. and E.R.S. provided tools used in the study. J.S.S., T.J.S.S., R.J.B., A.Z.K., E.F.S., K.M.D., S.E.N., and J.V.R. carried out the investigations. J.S.S., C.J.A., and J.V.R. wrote the manuscript. J.V.R. supervised the study, and funding acquisition was by C.J.A. and J.V.R.

## Competing interests
The authors declare no competing interests.
