## [Peer Review File · Nature Communications]

Reviewers' Comments:

Reviewer #1:

Remarks to the Author:

The manuscript by Selfe et al demonstrates an all-optical imaging strategy to compare GABA reversal potentials in organotypic cultures and in vivo. This approach uses a fusion protein of a genetically encoded light-activated anion channelrhodopsin and membrane voltage sensor to both activate Cl⁻ fluxes and measure corresponding changes in membrane potential, respectively. The study characterizes how the tool functions and how it can be employed in different cell types. The authors demonstrate the utility of this approach in several experimental paradigms that both validate previous works in the field, as well as provide novel insights into astrocyte chloride dynamics during epileptiform activity in culture. Overall, the study is rigorous and interesting, and brings to bear a much-needed new tool to study chloride dynamics in neurons, astrocytes and potentially other cell types. I think that this study would be an important contribution to the field, however I do think the manuscript would be strengthened by addressing the following:

Major:

- 1) Given the technical nature of the manuscript, more description of the methodology, specifically related to Voltron, should be detailed in the main text. Is there any normalization done or can be done for Voltron expression between cells for consistent intensimetric signals? Is the V_m-fluorescence relationship linear at potentials above -20mV? It looks like this might be approaching the upper limit of the dynamic range for voltron.
- 2) It is unclear how the DFGABA estimation is achieved in cells that were not patch-clamped (e.g. fig 2c&f and elsewhere). Are these estimations just extrapolations from patched neurons? How variable is F₀ between cells – and what does this value correspond to? For example, it seems like F₀ is set to -60mV, is this assuming that resting membrane potential is -60mV across neurons? Is there much variability in Voltron expression between cells? How these estimations are performed should be clearly described in the main text.
- 3) As an extension of the above point, an important validation of the tool's utility would be to measure the accuracy ORCHID DFGABA readouts in unpatched cells. The authors nicely show a linear relationship in Voltron fluorescence changes with GABA currents in patched cells. However, to demonstrate the true accuracy of ORCHID I suggest the authors optically determine DFGABA in an unpatched neuron (such as in Fig 2) and then compare this optical readout afterwards, in the same cell, using a perforated patch to measure EGABA.
- 4) The authors should discuss caveats/limitations of this approach in the context of existing imaging tools to measure intracellular chloride.

Minor:

5) In example Fig 4c it appears that depolarizing GABA currents are sufficient to trigger SLE-like events. How often does this occur? Is it only in interneurons, or pyramidal neurons as well?

6) An analysis of the temporal DFGABA changes leading up to and including the 'pre-SLE' to SLE (i.e. all readouts leading up to, including, and after the SLE) would be interesting to see.

7) There are a number of studies that have imaged high Cl⁻ in astrocytes, including some already cited (Untiet, Weilinger), that should be referenced in the appropriate sections. It would also be nice to reference early work on GABA-induced depolarization in astrocytes (e.g. Kettenmann and colleagues in the 80s). There are also several studies that have demonstrated KCC2/NKCC1 expression differences in neurons vs astrocytes that should be cited (related to fig 5f).

8) It would be interesting to see more in-depth analysis of the heterogeneity of the in vivo ORCHID readouts during basal activity (e.g. as in Fig 6b). Are there correlations between spontaneous field activity and GtACR2-Voltron readouts?

Reviewer #2:

Remarks to the Author:

In their manuscript entitled „All-optical reporting of inhibitory receptor driving force in the nervous system“ Joshua Selfe and coworkers described a new experimental approach to investigate one major aspect of GABAergic function, the driving force of the GABA_A-receptor (DF-GABA_A). As the DF-GABA_A is one essential parameter determining GABAergic inhibition and since a variety of studies described that the GABAergic driving force is influenced by a variety of physiological and pathophysiological conditions, the possibility to determine this parameter under minimal-invasive conditions with pre possibility of massive parallel recordings is advantageous. Therefore, the topic of this manuscript is highly relevant. The chosen approach is innovative and is adequately described, the results of the experiments are in most cases properly presented and illustrated. The discussion is in most parts fair and nicely emphasized future applications of the ORCHID technique.

However, besides some minor comments, I have a general problem with the main statement of manuscript that “ORCHID (all-Optical Reporting of Chloride Ion Driving force) as a new tool for quantifying DF-GABA_A”, as probably only limited, but important, information about DF-GABA_A can be obtained with this method. In my opinion the manuscript should be carefully and thoroughly rephrased to honestly describe the limitations of this ORCHID technique and to realistically suggest which important

questions can be investigated. In addition, the reliability of the main findings should be displayed in more detail in the initial parts of the results and the related figure. Despite this major dilemma, I like to state that the proposed technical approach will be most probable of high practical utility for researchers interested in the dynamics and development of GABAergic inhibition and will also open up new strategies to determine the functional consequences of ionic dynamics in the CNS.

Major point 1: Can the DF-GABA_A be determined by ORCHID?

The authors describe “ORCHID (all-Optical Reporting of Chloride Ion Driving force) as a new tool for quantifying DF-GABA_A”. (Line 22-24). The authors use the chemogenetic voltage-sensor “Voltron2-ST [...] to measure the magnitude and direction of GtACR2 anion-current-induced changes to V_m ”. However, the amplitude of the GtACR2-induced voltage changes depends not only on the DF-GABA_A, but also on the GtACR2-induced change in the membrane-conductance and the input-resistance of the cell. In addition, the authors do not provide sufficient information whether the absolute value of the membrane potential can be assumed from the DF/F₀ values, so in fact you can neither estimate the absolute voltage at the peak of the Df/F₀ response, nor the absolute current of the GtACR2 responses. In my opinion, it is therefore not possible to determine, or even estimate, the value of the DF-GABA_A just from the amplitude of the Voltron2-ST-based DF/F₀ signal. In my opinion this limitation of the ORCHID technique must be stated in more detail in the manuscript.

But in fact, the technique is perfectly suited, as demonstrated by the experiments in the present manuscript, to determine changes in the “DF-GABA_A polarity” (line 463). Therefore I would suggest, that the author may change their description focusing more exactly on this particular readout of their method. In addition, the authors provide nice data about the stability of the Voltron2-ST-based DF/F₀ signals and the GtACR2 responses. Therefore, I consider that the ORCHID technique is also perfectly suited to follow the development of relative changes on the DF-GABA_A, even if no switch in the polarity of the DF-GABA_A occurs. The authors should also discuss this aspect in much more detail.

However, even if this technique provides “only” the DF-GABA_A polarity, it opens an important avenue for the investigation of many highly relevant questions, as highlighted e.g. by the observation of stable outward-directed DF-GABA_A in astrocytes during massive activity transients (Line 481).

Major Point 2: How reliable can the DF-GABA_A be determined.

In Fig. 1n most probably only the values of the typical recording in Fig 1m are plotted. On

the other hand, Fig 1l as well as the SEM-error bars in Fig. 1j suggests a rather large scatter in the absolute DF/D0 values between single cells. As the key message of your manuscript is, that ORCHID can be used to quantify the DF-GABA_A I would strongly suggest to include in this graph the individual datapoints for all experiments to allow the reader to judge on the reliability of the DF-GABA_A determination by the fluorescence signals (e.g. variability in the slope and the reversal of the regression line).

In addition, for Fig. 1n (and Suppl. Fig. 4) you probably have for each cell a voltage to DF/D0 calibration. For the suggested purpose it is, however, necessary to estimate the accuracy of DF-GABA_A determination with your method if a patch-related internal voltage calibration was not available. Is it from your datasets with this simultaneous DEVI and patch-clamp recordings possible to quantify to which degree the amplitude and/or polarity of DF-GABA_A-CC can be directly and cell-independently predicted from the DF-GABA_A-Fluoro.

With respect to my first major point, can you quantify the relative stability of DF-GABA_A-Fluoro values along the 45 min recording interval (Fig. 1l)? Here proving only an ANOVA demonstrating the absence of a difference is in my opinion not sufficient.

Please also extend these analyses to the high chloride condition, as displayed in Supl. Fig 2. Do you have also simultaneous DF-GABA_A-CC and DF-GABA_A-Fluo measurements using a high-chloride pipette solution? Can you state whether the reversal potentials determined from IV plot using DEVI signals are consistent with the reversal potentials estimated by the chloride concentration in the pipette?

Finally, a quantification of the reliability of the DF-GABA_A polarity determination between independent measurements/experiments may support your analysis. You may consider to estimate from a I-V-plot the reversal potential from the linear regression of either GEVI or patch-clamp data, and quantify the difference/similarity between both values.

Minor

Line 18: Please consider to refer to the more actual comprehensive review by Farrant & Kaila (2007).

Line 36: Regarding the diurnal chloride changes in neurons I would suggest to include a reference to the seminal Wagner and Yarom (1997) publication on this topic.

Line 57: Please discuss, whether you can neglect the Voltron2-St induced

photocurrents in your recordings.

Line 61: Please consider to slightly tone down the statement “This means that they share the same relative permeability to Cl⁻ and HCO₃⁻” as to my knowledge this has not been demonstrated, but it extremely plausible.

Line 106: Please note that in Fig 1h the GEVI and patch traces are not accurately aligned.

Fig. 3h: This seems to be the only traces in which the noise increases this much during the recording. Regarding the fact that you prove the stability of the DF/F0 signal during long recordings, I wonder why you illustrate the VU effect with particular this recording. The time course of the washout trace (which suggest a prominent adaptation and a rebound effect) makes me wonder how you quantified this response.

Line 267, Fig. 4c: While the SLE induced shift in the DF-GABA_A polarity was nicely illustrated by the phase-locked induction of clonic-tonic like bursts upon GtACR2 activation, the complex Voltron2-ST fluorescence signal, probably contaminated by spikes, raises the question whether this voltage signal reflects only the GtACR2-based potential shifts and are thus suited for a determination of DF-GABA_A.

Line 378: Here I would suggest to avoid this differentiation between “hyperpolarizing, shunting and hyperpolarizing DF-GABA_A”. There are several reports that depolarizing GABA responses, related to a positive DF-GABA_A, can also mediate shunting effects. Probably the term “purely shunting” would be more accurate.

Line 417: “ ORCHID’s accuracy in reporting DFGABAA depends upon the amplitude of the light activated anion conductance, relative to other conductances that are active in the cell of interest.” Please specify this information. Is this just related to the decreases signal to noise ratio? Or do you implicate a dominance of the GtARC2 conductance that clamps E_m to the GABA reversal potential is a necessary prerequisite to determine the DF-GABA_A?

Line 508: Here I would suggest to also tone down this suggestion. The case of DF-GABA_A is special, as there a shift in the polarity can occur due to the fact that the Cl⁻ concentration is close to a passive distribution. For other physiologically relevant ions like Na⁺ or Ca²⁺ (with the exception of HCO₃⁻) one may assume that no such large changes in the respective reversal potential can be observed, as here the ion gradients are so massive. On the other hand, you may speculate whether an optical technique like ORKID is feasible to investigate alterations in the extracellular K⁺ concentration.

Reviewer #3:

Remarks to the Author:

Selze and colleagues introduce a novel combination of tools for quantifying the inhibitory receptor driving force, which they have named ORCHID: all-Optical Reporting of Chloride Ion Driving force. ORCHID substantiates theoretical predictions regarding the biophysical mechanisms underlying DFGABAA and provides the inaugural in vivo measurements of intact DFGABAA. This contribution significantly advances our comprehension of inhibitory synaptic transmission and sets a new standard for employing all-optical methodologies in the assessment of ionic driving forces. The tool presented by Selze et al. is exceedingly promising and holds considerable interest for the rapidly expanding field of GABA and inhibition research, particularly for in vivo studies and in cells such as astrocytes, which have traditionally been challenging to record using electrophysiology in vivo. Nevertheless, the manuscript primarily focuses on introducing this method, and the authors do not sufficiently address the physiological and functional significance of their experimental findings.

Major:

- 1) Figure 1, and the corresponding results section is a crucial part of the manuscript which should engage the reader and make it easy to understand the basic concept of the presented tool. Therefore, figure 1b should be described in the results section not only in the figure legend. Figure 1a and c should be combined. Figure 1f, needs clarification, what do the illustrated traces show? Figure 1f, k, m, why is the DF/F_0 plotted as -%? Information about the fluorescent properties of the tool is missing. What is the reason for using different cell types in Figure 1a-f and Figure 1g onwards? Page 5, line 80-90, this paragraph is very technical and hard to follow. There are three different names for Voltron2-ST, GEVI Voltron2, GEVI. Be more consistent.
- 2) Figure 1 and figure 2, light activation of GtARC2 is a crucial part of the underlying mechanism of the presented tool. While the authors show that the activation of GtARC2 was stable over 45 min of repeated stimulation, a titration of this stimulation is missing. The information about the exact stimulation time is missing in all figures, and no information arguing for the used activation time is provided. Furthermore, a control or a reference is missing, which discusses the impact of repeated GtARC2 activation on neuronal baseline activity and especially during SLE. How does the short anion current upon GABA puff compare to the long current upon light stimulation? Would a shorter stimulation be less invasive? Why does Figure 1k illustrate a different cell type compared to Figure 1e and f. In which cell type has the repeated stimulation been performed?
- 3) Page 17, line 461-462, The authors claim that "By avoiding intracellular recordings, ORCHID enabled us to monitor unperturbed resting and dynamic DFGABAA." To make

this claim the authors need to address the impact of GtACR2 activation on baseline neuronal and astrocytic activity. The activation used in the presented manuscript is longer than GABA puffs. I suggest a statistical comparison of Figure 2c and f, as well as a similar comparison for recordings in astrocytes. Furthermore, repeated activation should be tested in all cell types presented in the paper.

4) Figure 3, the figure does not fit the stream of information presented in the results section. Different scales are used on all axis, which makes it difficult to compare the data. All shortcuts should be introduced in the text. Please introduce NHE. Figure 3j-m, has this approach been used and validated before? A reference is needed.

5) Figure 4, data from pyramidal and GAD2+ cells are shown, f-i are missing information which cell type this data is obtained from. In Fig. 4b, d: it is not obvious how these values are quantified; it is suggested that the corresponding periods (Baseline, Pre-SLE, Post-SLE, washout) be marked in the example traces (in 4a, c) and the process for quantification should be clearly described in the results and/or method section. The approach allows for a higher resolution than bulk averages, why are not the single recordings plotted over time?

6) Figure 4, and Figure 6 show DFGABAA from GAD2+ recorded from slices or in vivo. While in slices DFGABAA is always negative during baseline conditions, in vivo the baseline fluctuates around zero. Why are these baseline values so different? The same is true for values determined after 4-AP application.

7) Figure 5, Figure 5d and e would benefit from more detailed analysis. It is clear in d that there is a huge efflux during the first blue light stimulus and that it becomes much smaller during the later stimulations. Please split the “during SLE” average shown in Figure 5e into first and second stimulus or plot single stimulations over time. Fig. 5d: How do the authors explain the strong initial increase in fluorescence at the start of SLE? This should be noted and discussed in the text.

8) Figure 5, activation of GtACR2 should be titrated in astrocytes as well. Furthermore, controls should be provided to characterize the impact of the optogenetic tool on astrocytic physiology. This is crucial to highlight the limitations of the tool. Fig. 5b: Did the authors expose astrocytes to longer duration of optogenetic activation? Does the depolarization continue or reach a plateau? Is the ORCHID method applicable to astrocytes if there is no unambiguous maximum?

9) The authors need to make clear in every section of the results, which preparation was used, brain slices or in vivo. Furthermore, the information about the number of animals, slices and cells should be included in all figure legends. All symbols should be explained, and it should be obvious in the figures which cell type has been recorded from. Please also provide information about the stimulation paradigm with the corresponding data. Specifically, figure 1f, what do the two different traces show? Is it both fluorescence, one representative trace and one averaged trace? Figure 3, g, i, m: do grey symbols represent single or (repeated) averaged measures? Fig. 5e: Again, what do the symbols represent? How are the individual phases defined?

10) Immunohistochemistry is needed to confirm targeted expression of all constructs used.

11) Page 17, line 444-446, The authors claim that “detecting [Cl⁻]_i changes would require much larger and longer conductances that alter [Cl⁻]_i in the cytoplasm” this does not make sense regarding Cl imaging. Please clarify.

Minor:

1) Fig. 3 (line 228), what is gKCC2?

2) The used constructs (Voltron-2 and GtACR2) should be referenced in the main text.

3) The reference list needs to be checked carefully. Some citations are listed twice.

References for the main text and method section should be merged.

4) Please add Kaneko 2004 DOI:10.1523/JNEUROSCI.2115-04.2004 as reference for chloride FLIM. Page 4, line 48.

5) Page 5, line 88-90, more information about the reason for using this dye is needed. It was mentioned later, but it causes confusion in the beginning of the result section. Please rearrange the text.

6) Page 7, line 151-152, Here the construct is called ORCHID even though it is the Voltron only. I recommend keeping a clear nomenclature for all constructs used, to help the reader follow the experiments.

7) Figure 2, please provide information why different wavelengths were used to excite Voltron.

8) Using the same scale on 2b and c, as on 2e and f would help to compare the collected data.

9) Figure 2e, how long did the blue light stimulation last and why did you choose this duration?

10) There is a huge literature on NKCC1 in neurons and astrocytes, also transcriptomic data is available. Please include references and context based on previous publications.

11) Page 17, line 481, are able maintain, “to” is missing.

12) Methods, how were the mice killed?

13) Please provide a reference for DIV7-14 being equivalent to P14-21

Response to reviewers

Reviewer #1 (Remarks to the Author):

The manuscript by Selfe et al demonstrates an all-optical imaging strategy to compare GABA reversal potentials in organotypic cultures and in vivo. This approach uses a fusion protein of a genetically encoded light-activated anion channelrhodopsin and membrane voltage sensor to both activate Cl⁻ fluxes and measure corresponding changes in membrane potential, respectively. The study characterizes how the tool functions and how it can be employed in different cell types. The authors demonstrate the utility of this approach in several experimental paradigms that both validate previous works in the field, as well as provide novel insights into astrocyte chloride dynamics during epileptiform activity in culture. Overall, the study is rigorous and interesting, and brings to bear a much-needed new tool to study chloride dynamics in neurons, astrocytes and potentially other cell types. I think that this study would be an important contribution to the field, however I do think the manuscript would be strengthened by addressing the following:

Major:

1) Given the technical nature of the manuscript, more description of the methodology, specifically related to Voltron, should be detailed in the main text. Is there any normalization done or can be done for Voltron expression between cells for consistent intensimetric signals? Is the V_m-fluorescence relationship linear at potentials above -20mV? It looks like this might be approaching the upper limit of the dynamic range for voltron.

On the reviewer's suggestion we have now added more technical detail regarding methodology, particularly related to Voltron2-ST imaging, both in the Results and the Methods sections.

To normalize for Voltron2-ST expression levels and to improve the consistency of intensimetric fluorescence measurements, the $\Delta F/F_0$ measurement was used, where F_0 is the baseline fluorescence intensity and ΔF is the magnitude of the change to F_0 . This is a typical normalization method utilised with GEVI and GECI imaging of this nature. No other normalization method was employed.

The fluorescence-voltage relationship of Voltron2-ST has been investigated before. See Abdelfattah et al., (2023). Here we have focused particularly on the voltage changes relevant to the measurement of DF_{GABAA} , with cells being voltage clamped at -60 mV, and recordings made of 10 mV voltage steps from -100 mV to -30 mV, in this range the fluorescence-voltage relationship is linear (see Fig. 1j).

2) It is unclear how the DF_{GABA} estimation is achieved in cells that were not patch-clamped (e.g. fig 2c&f and elsewhere). Are these estimations just extrapolations from patched neurons? How variable is F_0 between cells – and what does this value correspond to? For example, it seems like F_0 is set to -60mV, is this assuming that resting membrane potential is -60mV across neurons? Is there much variability in Voltron expression between cells? How these estimations are performed should be clearly described in the main text.

We have clarified this with new text in the Results and Methods sections, explaining that using the relationship between $\Delta F/F_0$ and ΔV for Voltron2₆₀₈-ST and for Voltron2₅₄₉-ST, which was acquired and averaged from multiple simultaneously patched and imaged neurons (see new Fig. 1j), DF_{GABAA} could be estimated from cells which were imaged alone. The reviewer is thus correct that DF_{GABAA} estimations are extrapolations from patched neurons.

As in all GEVI or GECI imaging, F_0 was highly variable between cells depending on Voltron2-ST expression levels, dye level, the depth of the imaged cells etc. This was mitigated by using $\Delta F/F_0$ as a measure of fluorescence change for our analysis. F_0 was not set to -60 mV or any other value. Rather, given that the fluorescence-voltage relationship (i.e., ΔV vs $\Delta F/F_0$) is linear (see Fig. 1j), a measured $\Delta F/F_0$ in cells imaged alone could be translated to a ΔV as previously performed by Cornejo et al., (2022). This translated ΔV was then used to estimate DF_{GABAA} . We now more thoroughly and clearly describe this using new text and figures in the Results and Methods sections.

3) As an extension of the above point, an important validation of the tool's utility would be to measure the accuracy ORCHID DF_{GABA} readouts in unpatched cells. The authors nicely show a linear relationship in Voltron fluorescence changes with GABA currents in patched cells. However, to demonstrate the true accuracy of ORCHID I suggest the authors optically determine DF_{GABA} in an unpatched neuron (such as in Fig 2) and then compare this optical readout afterwards, in the same cell, using a perforated patch to measure EGABA.

On the reviewer's suggestion we now include data from several ORCHID-expressing neurons which were each imaged concurrent with a gramicidin-perforated patch-clamp recording (Fig. 1n; Supplementary Fig. 2a). We show that optically measured DF_{GABAA} using ORCHID was equivalent to the DF_{GABAA} calculated in current-clamp mode from the gramicidin-perforated patch-clamp recording. In addition, we now also provide data of DF_{GABAA} measured using gramicidin-perforated patch-clamp recordings from a population of pyramidal neurons in the same preparation (hippocampal organotypic brain slices) to show that these are comparable to the optical DF_{GABAA} estimates we measured using ORCHID (Supplementary Fig. 2b).

4) The authors should discuss caveats/limitations of this approach in the context of existing imaging tools to measure intracellular chloride.

We have now added a section to the discussion where we discuss the limitations of ORCHID in the context of existing imaging tools to measure intracellular chloride. We state that ORCHID, whilst it can provide an estimate of DF_{GABAA} , cannot provide a measurement of absolute intracellular Cl^- concentration nor E_{GABAA} . In addition, it cannot provide an absolute estimate of V_m .

Minor:

5) In example Fig 4c it appears that depolarizing GABA currents are sufficient to trigger SLE-like events. How often does this occur? Is it only in interneurons, or pyramidal neurons as well?

The reviewer is correct that GtACR2 activation during ORCHID recordings from GAD2+ interneurons after an SLE often caused further network bursts to occur (Fig. 4c). This occurred in 8/12 GAD2+ interneuronal recordings, and not when recording from CaMKII α + pyramidal neurons (0/15). This analysis and an associated new supplementary figure (Supplementary Fig. 10) have been added to the manuscript.

6) An analysis of the temporal DF_{GABA} changes leading up to and including the 'pre-SLE' to SLE (i.e. all readouts leading up to, including, and after the SLE) would be interesting to see.

This analysis is now provided in Supplementary Fig. 9 for neurons and Supplementary Fig. 12 for astrocytes.

7) There are a number of studies that have imaged high Cl⁻ in astrocytes, including some already cited (Untiet, Weilinger), that should be referenced in the appropriate sections. It would also be nice to reference early work on GABA-induced depolarization in astrocytes (e.g. Kettenmann and colleagues in the 80s). There are also several studies that have demonstrated KCC2/NKCC1 expression differences in neurons vs astrocytes that should be cited (related to fig 5f).

We thank the reviewer for these useful recommendations. Citations to these studies have been added where appropriate.

8) It would be interesting to see more in-depth analysis of the heterogeneity of the in vivo ORCHID readouts during basal activity (e.g. as in Fig 6b). Are there correlations between spontaneous field activity and GtACR2-Voltron readouts?

We agree with the reviewer that further in-depth analysis of the heterogeneity of the *in vivo* ORCHID readouts during basal activity would be interesting. Although we have not performed more in-depth analysis, we have added sentences to the discussion on this excellent point: “Our *in vivo* results captured a range of resting DF_{GABAA} values in cortical L1 GAD2+ interneurons, which is different to what was observed *in vitro*. We hypothesize that this heterogeneity may be due, in part, to differences in network state (Burman *et al.*, 2023) or the animal’s sleep-wake history (Alfonsa *et al.*, 2023).”

Reviewer #2 (Remarks to the Author):

In their manuscript entitled “All-optical reporting of inhibitory receptor driving force in the nervous system” Joshua Selfe and coworkers described a new experimental approach to investigate one major aspect of GABAergic function, the driving force of the GABA_A-receptor (DF-GABA_A). As the DF-GABA_A is one essential parameter determining GABAergic inhibition and since a variety of studies described that the GABAergic driving force is influenced by a variety of physiological and pathophysiological conditions, the possibility to determine this parameter under minimal-invasive conditions with the possibility of massive parallel recordings is advantageous. Therefore, the topic of this manuscript is highly relevant. The chosen approach is innovative and is adequately described, the results of the experiments are in most cases properly presented and illustrated. The discussion is in most parts fair and nicely emphasized future applications of the ORCHID technique.

However, besides some minor comments, I have a general problem with the main statement of manuscript that “ORCHID (all-Optical Reporting of Chloride Ion Driving force) as a new tool for quantifying DF-GABA_A”, as probably only limited, but important, information about DF-GABA_A can be obtained with this method. In my opinion the manuscript should be carefully and thoroughly rephrased to honestly describe the limitations of this ORCHID technique and to realistically suggest which important questions can be investigated. In addition, the reliability of the main findings should be displayed in more detail in the initial parts of the results and the related figure. Despite this major dilemma, I like to state that the proposed technical approach will be most probable of high practical utility for researchers interested in the dynamics and development of GABAergic inhibition and will also open up new strategies to determine the functional consequences of ionic dynamics in the CNS.

Major point 1: Can the DF-GABA_A be determined by ORCHID?

The authors describe “ORCHID (all-Optical Reporting of Chloride Ion Driving force) as a new tool for quantifying DF-GABA_A”. (Line 22-24). The authors use the chemogenetic voltage-sensor “Voltron2-ST [...] to measure the magnitude and direction of GtACR2 anion-current-induced changes to V_m”. However, the amplitude of the GtACR2-induced voltage changes depends not only on the DF-GABA_A, but also on the GtACR2-induced change in the membrane-conductance and the input-resistance of the cell.

We thank the reviewer for emphasizing this point. The reviewer is quite correct that “the amplitude of the GtACR2-induced voltage changes depends not only on the DF-GABA_A, but also on the GtACR2-induced change in the membrane-conductance and the input-resistance of the cell”. We now state in the discussion that “ORCHID’s accuracy in reporting DF_{GABAA} depends upon the size of the light-activated anion conductance (the GtACR2-induced change in the membrane conductance) relative to other conductances that are active in the cell of interest”. That is, the bigger the GtACR2-induced change in the membrane conductance relative to other conductances in the cell, the more accurate the DF_{GABAA} estimate. We point to the fact that GtACR2 is known to generate very large conductances and that any future optimization of light-activated anion channels are predicted to further improve ORCHID’s accuracy.

In addition, the authors do not provide sufficient information whether the absolute value of the membrane potential can be assumed from the DF/F₀ values, so in fact you can neither estimate the absolute voltage at the peak of the Df/F₀ response, nor the absolute current of the GtACR2 responses.

We agree and now state in the manuscript (see Discussion) that the absolute value of the membrane potential cannot be assumed from $\Delta F/F_0$ values. However, as we demonstrate in a new Fig. 1j using recordings from multiple simultaneously patched and imaged cells, the relationship between a change in membrane voltage (ΔV) and $\Delta F/F_0$ is linear. Therefore, in cells that are only imaged, $\Delta F/F_0$ can be used to calculate a likely change in membrane voltage that caused the $\Delta F/F_0$. This technique has been used previously, notably by Cornejo et al., (2022). This change in membrane voltage is then used to estimate DF_{GABAA}. We do not claim that Voltron2-ST imaging can be used to estimate the absolute current of the GtACR2 response, nor do we believe this as necessary for ORCHID to provide an optical estimate of DF_{GABAA}.

In my opinion, it is therefore not possible to determine, or even estimate, the value of the DF-GABA_A just from the amplitude of the Voltron2-ST-based DF/F₀ signal. In my opinion this limitation of the ORCHID technique must be stated in more detail in the manuscript.

For the reasons described above, we believe that a useful estimate of DF_{GABAA} can be generated from the amplitude and direction of the Voltron2-ST-based $\Delta F/F_0$ signal. We thank the reviewer for encouraging us to be clearer about important considerations regarding the ORCHID approach. We now provide much more detail in the Results and Methods sections for how DF_{GABAA} is estimated, and in addition we are clearer about the limitations of the ORCHID technique (see revisions to the Discussion).

But in fact, the technique is perfectly suited, as demonstrated by the experiments in the present manuscript, to determine changes in the “DF-GABA_A polarity” (line 463). Therefore I would suggest, that the author may change their description focusing more exactly on this

particular readout of their method. In addition, the authors provide nice data about the stability of the Voltron2-ST-based DF/F₀ signals and the GtACR2 responses. Therefore, I consider that the ORCHID technique is also perfectly suited to follow the development of relative changes on the DF-GABA_A, even if no switch in the polarity of the DF-GABA_A occurs. The authors should also discuss this aspect in much more detail.

We thank the reviewer for their enthusiasm that the ORCHID technique is particularly suited for demonstrating the DF_{GABA_A} polarity as well as relative changes in DF_{GABA_A}. We agree with the reviewer and have highlighted this aspect of ORCHID in the revised Discussion section.

However, even if this technique provides “only” the DF-GABA_A polarity, it opens an important avenue for the investigation of many highly relevant question, as highlighted e.g. by the observation of stable outward-directed DF-GABA_A in astrocytes during massive activity transients (Line 481).

We thank the reviewer for highlighting the utility of ORCHID in this regard.

Major Point 2: How reliable can the DF-GABA_A be determined.

In Fig. 1n most probably only the values of the typical recording in Fig 1m are plotted. On the other hand, Fig 1l as well as the SEM-error bars in Fig. 1j suggests a rather large scatter in the absolute DF/D₀ values between single cells. As the key message of your manuscript is, that ORCHID can be used to quantify the DF-GABA_A I would strongly suggest to include in this graph the individual datapoints for all experiments to allow the reader to judge on the reliability of the DF-GABA_A determination by the fluorescence signals (e.g. variability in the slope and the reversal of the regression line).

We agree with the reviewer and have now substantially revised Fig. 1j (and Supplementary Fig. 4) to address this point. We now plot the ΔV versus $\Delta F/F_0$ relationship for each recorded cell independently as well as the mean, to give the reader a better sense of the variability in the slope and the reversal of the regression line between cells. Linear regression results in a slope of 0.1845 ± 0.01 , which equates to 5.42 mV per $\Delta F/F_0$ percent ($R^2 = 0.9311$, $n = 7$ cells).

In addition, for Fig. 1n (and Suppl. Fig. 4) you probably have for each cell a voltage to DF/D₀ calibration. For the suggested purpose it is, however, necessary to estimate the accuracy of DF-GABA_A determination with your method if a patch-related internal voltage calibration was not available. Is it from your datasets with this simultaneous DEVI and patch-clamp recordings possible to quantify to which degree the amplitude and/or polarity of DF-GABA_A-CC can be directly and cell-independently predicted from the DF-GABA_A-Fluoro.

As suggested by the reviewer, in Fig. 1n (and Supplementary Fig. 5d), we now plot data points from individual cells, which were simultaneously patched and imaged to allow the reader “to quantify to which degree the amplitude and/or polarity of DF-GABA_A-CC can be directly and cell-independently predicted from the DF-GABA_A-Fluoro”. We have also added statistical analysis of the regression to the Results section. With $R^2 = 0.9801$ (Runs test, deviation from linearity: $P = 0.9360$, $n = 24$ recordings from 17 cells) this indicates an excellent fit of the regression model to our data.

With respect to my first major point, can you quantify the relative stability of DF-GABA_A-

Fluoro values along the 45 min recording interval (Fig. 1l)? Here proving only an ANOVA demonstrating the absence of a difference is in my opinion not sufficient.

We have now added additional statistical analysis to quantify the relative stability of DF_{GABAA} estimates for Fig. 1l, and Supplementary Fig. 5b. This can be found in the respective figure legends as well as in Supplementary Table 1 and 2. In summary we found no statistical difference between any pair of means at any time point confirming the stability of the recordings.

Please also extend these analyses to the high chloride condition, as displayed in Supl. Fig 2. Do you have also simultaneous DF-GABAa-CC and DF-GABAa-Fluo measurements using a high-chloride pipette solution? Can you state whether the reversal potentials determined from IV plot using DEVI signals are consistent with the reversal potentials estimated by the chloride concentration in the pipette?

As requested by the reviewer we have now added data of simultaneous $DF_{GABAA-CC}$ and $DF_{GABAA-Fluo}$ measurements using a high-chloride pipette solution to Fig. 1n, as well as Supplementary Fig. 5d. As explained previously, it is not possible to determine reversal potentials (E_{GABAA}) using ORCHID as the GEVI signal reports relative voltage change and not absolute voltage. That is, we can estimate DF_{GABAA} , but without a measurement of absolute voltage we cannot estimate E_{GABAA} nor compare this to the expected E_{GABAA} given the Cl^- concentration of the pipette.

Finally, a quantification of the reliability of the DF-GABAa polarity determination between independent measurements/experiments may support your analysis. You may consider to estimate from a I-V-plot the reversal potential from the linear regression of either GEVI or patch-clamp data, and quantify the difference/similarity between both values.

Although we are unable to estimate reversal potentials as explained above, we were able to quantify the reliability of DF_{GABAA} estimates between independent measures (see Fig. 1n and Supplementary Fig. 5d). With a $R^2 = 0.9801$ (Runs test, deviation from linearity: $P = 0.9360$, $n = 24$ recordings from 17 cells) we were able to demonstrate the reliability of our DF_{GABAA} estimates using ORCHID.

Minor

Line 18: Please consider to refer to the more actual comprehensive review by Farrant & Kaila (2007).

This pertinent reference has been added as suggested.

Line 36: Regarding the diurnal chloride changes in neurons I would suggest to include a reference to the seminal Wagner and Yarom (1997) publication on this topic.

This reference has been added as suggested.

Line 57: Please discuss, whether you can neglect the Voltron2-St induced photocurrents in your recordings.

We have now added the following to the first section of the Results: “steady-state illumination produces negligible photocurrents by Voltron2-ST itself (Abdelfattah *et al.*, 2023)”.

Line 61: Please consider to slightly tone down the statement “This means that they share the same relative permeability to Cl⁻ and HCO₃⁻” as to my knowledge this has not been demonstrated, but it extremely plausible.

We have now toned down this statement to read as follows: “This suggests that they likely share the same relative permeability to Cl⁻ and HCO₃⁻.”

Line 106: Please note that in Fig 1h the GEVI and patch traces are not accurately aligned.

We apologize for this error. The traces are now accurately aligned.

Fig. 3h: This seems to be the only traces in which the noise increases this much during the recording. Regarding the fact that you prove the stability of the DF/F₀ signal during long recordings, I wonder why you illustrate the VU effect with particular this recording. The time course of the washout trace (which suggest a prominent adaptation and a rebound effect) makes me wonder how you quantified this response.

We have gathered additional VU data for Fig. 3h and now utilise an alternative example with no change in noise to illustrate the VU effect. Additionally, a more detailed explanation of how the response is quantified is given in the Methods section (Data Analysis and Statistics).

Line 267, Fig. 4c: While the SLE induced shift in the DF-GABA_A polarity was nicely illustrated by the phase-locked induction of clonic-tonic like bursts upon GtACR2 activation, the complex Voltron2-ST fluorescence signal, probably contaminated by spikes, raises the question whether this voltage signal reflects only the GtACR2-based potential shifts and are thus suited for a determination of DF-GABA_A.

Thank you for emphasizing this important point. We do not record DF_{GABA_A} during the clonic-tonic like bursts, as we agree that this signal is unsuitable for determination of DF_{GABA_A} (it is contaminated by spikes and/or the GtACR2 conductance increase is overwhelmed by other concurrent conductances). We have now added additional figures (Supplementary Fig. 9 and Supplementary Fig. 10) to clarify this issue. We show that GtACR2 activation during ORCHID recordings from GAD2⁺ interneurons after an SLE caused further clonic-tonic like bursts to occur (Fig. 4c) in 8 of 12 GAD2⁺ interneuronal recordings (Supplementary Fig. 10). In these recordings a post-SLE estimate of DF_{GABA_A} (Fig. 4d) was not collected (dashed lines).

Line 378: Here I would suggest to avoid this differentiation between “hyperpolarizing, shunting and hyperpolarizing DF-GABA_A”. There are several reports that depolarizing GABA responses, related to a positive DF-GABA_A, can also mediate shunting effects. Probably the term “purely shunting” would be more accurate.

This has been changed as suggested.

Line 417: “ ORCHID’s accuracy in reporting DFGABA_A depends upon the amplitude of the light activated anion conductance, relative to other conductances that are active in the cell of interest.” Please specify this information. Is this just related to the decreases signal to noise ratio? Or do you implicate a dominance of the GtARC2 conductance that clamps E_m to the GABA reversal potential is a necessary prerequisite to determine the DF-GABA_A?

The latter is correct, “a dominance of the GtARC2 conductance that clamps E_m to the GABA reversal potential is a necessary prerequisite” to estimate DF_{GABAA} . This is now explained more thoroughly in the Results and the Methods sections. Furthermore, the limitations of this approach are highlighted in the Discussion, including in the sentence quoted above.

Line 508: Here I would suggest to also tone down this suggestion. The case of DF-GABA_a is special, as there a shift in the polarity can occur due to the fact that the Cl⁻ concentration is close to a passive distribution. For other physiologically relevant ions like Na⁺ or Ca²⁺ (with the exception of HCO₃⁻) one may assume that no such large changes in the respective reversal potential can be observed, as here the ion gradients are so massive. On the other hand, you may speculate whether an optical technique like ORKID is feasible to investigate alterations in the extracellular K⁺ concentration.

The reviewer’s point is well taken, and we agree that the strategy we propose will be more or less suitable to assess ionic-driving forces depending on the particular context. Additionally, the reviewer’s suggestion of “ORKID” to assess how extracellular K⁺ changes could affect the driving force for K⁺ is an excellent example of how our strategy could be used to investigate alternative ionic driving force changes beyond that of GABA_ARs or Cl⁻. We therefore respectfully request that our concluding sentence remain as is.

Reviewer #3 (Remarks to the Author):

Selfe and colleagues introduce a novel combination of tools for quantifying the inhibitory receptor driving force, which they have named ORCHID: all-Optical Reporting of Chloride Ion Driving force. ORCHID substantiates theoretical predictions regarding the biophysical mechanisms underlying DFGABAA and provides the inaugural in vivo measurements of intact DFGABAA. This contribution significantly advances our comprehension of inhibitory synaptic transmission and sets a new standard for employing all-optical methodologies in the assessment of ionic driving forces. The tool presented by Selfe et al. is exceedingly promising and holds considerable interest for the rapidly expanding field of GABA and inhibition research, particularly for in vivo studies and in cells such as astrocytes, which have traditionally been challenging to record using electrophysiology in vivo. Nevertheless, the manuscript primarily focuses on introducing this method, and the authors do not sufficiently address the physiological and functional significance of their experimental findings.

Major:

1) Figure 1, and the corresponding results section is a crucial part of the manuscript which should engage the reader and make it easy to understand the basic concept of the presented tool. Therefore, figure 1b should be described in the results section not only in the figure legend. Figure 1a and c should be combined. Figure 1f, needs clarification, what do the illustrated traces show? Figure 1f, k, m, why is the DF/F0 plotted as -%? Information about the fluorescent properties of the tool is missing. What is the reason for using different cell types in Figure 1a-f and Figure 1g onwards? Page 5, line 80-90, this paragraph is very technical and hard to follow. There are three different names for Voltron2-ST, GEVI Voltron2, GEVI. Be more consistent.

We thank the reviewer for the point regarding Fig. 1b, which we now refer to in the main text of the Results section. However, we respectfully request to keep Fig. 1a and Fig. 1c separate, as they refer to related but distinct ideas. Fig. 1a refers to the conceptual design

for how an all-optical strategy to estimate DF_{GABA_A} would work. Fig. 1c depicts how this strategy is actually implemented by ORCHID. Regarding Fig. 1f, we have added the following explanation to the figure legend: "... no change in measured fluorescence in either a recording from a single cell (top, orange trace), or in the averaged recorded fluorescence from 29 neurons (showing the mean, orange trace \pm standard error of the mean, SEM, grey shading)." We hope this clarifies what the traces show. Furthermore, we have added to the Results an explanation for why $\Delta F/F_0$ is plotted as a negative: "Voltron2-ST fluorescence readout ($\Delta F/F_0$) was plotted as a negative for intuitive understanding of the underlying V_m shifts, as Voltron2-ST fluorescence increases with a decrease in V_m ." We have also added a more thorough description of the fluorescent properties of the tool to the Results and have improved the readability of the paragraph in the original "Page 5, line 80-90". The reason for the different cell types displayed in Fig. 1 is that the cell type displayed reflects the cell type from which each specific example trace was recorded. Finally, as suggested by the Reviewer we have now changed the text to be more consistent regarding the naming of Voltron2-ST.

2) Figure 1 and figure 2, light activation of GtARC2 is a crucial part of the underlying mechanism of the presented tool. While the authors show that the activation of GtARC2 was stable over 45 min of repeated stimulation, a titration of this stimulation is missing. The information about the exact stimulation time is missing in all figures, and no information arguing for the used activation time is provided. Furthermore, a control or a reference is missing, which discusses the impact of repeated GtARC2 activation on neuronal baseline activity and especially during SLE. How does the short anion current upon GABA puff compare to the long current upon light stimulation? Would a shorter stimulation be less invasive? Why does Figure 1k illustrate a different cell type compared to Figure 1e and f. In which cell type has the repeated stimulation been performed?

We do agree that light activation of GtACR2 is a crucial part of the presented tool. A titration of stimulation light intensity and the resulting GtACR2 activation is shown in Fig. 1d. We now clearly state the stimulation time where appropriate in the figures and the main text of the Results. Furthermore, a thorough explanation and derivation for the chosen activation time based on the targeted signal to noise ratio is now provided in the Methods (see Widefield Fluorescence Imaging). We have added a reference to the discussion regarding the impact of repeated GtACR2 activation, and have provided a rationale for the similarity between the short anion current upon GABA puffs and the long anion current upon light stimulation (see Methods, Widefield Fluorescence Imaging). Regarding the chosen duration of the GABA puff, we have added: "This duration was chosen to maximize GABA delivery while minimizing tissue movement." Again, we now explain that the duration of ORCHID stimulation was chosen to maximise the signal-to-noise ratio (see Methods, Widefield Fluorescence Imaging for calculation). Finally, we rationalise that: "These durations (*of ORCHID stimulation*) are broadly similar to the duration of GABA_AR stimulation during micropipette delivery of GABA." In terms of the invasiveness of the optogenetic stimulation, optogenetic stimulation is generally minimally invasive at brief stimulation times, such as we employ (5 x 2 s typically) (Wiegert *et al.*, 2017). Please see the new Supplementary Fig. 15 for data confirming this in astrocytes. Regarding the different cell types, Fig. 1k illustrates a different cell type compared to Fig. 1e and Fig. 1f because the cell type displayed reflects the cell type from which each specific example trace was recorded. In Fig. 1l, the population data includes recordings from both pyramidal neurons (n = 7) and interneurons (n = 7) (the example trace is from an interneuron).

3) Page 17, line 461-462, The authors claim that “By avoiding intracellular recordings, ORCHID enabled us to monitor unperturbed resting and dynamic DF_{GABAA} .” To make this claim the authors need to address the impact of GtACR2 activation on baseline neuronal and astrocytic activity. The activation used in the presented manuscript is longer than GABA puffs. I suggest a statistical comparison of Figure 2c and f, as well as a similar comparison for recordings in astrocytes. Furthermore, repeated activation should be tested in all cell types presented in the paper.

Please see the additional text added to the discussion for references regarding the impact of GtACR2 activation on baseline neuronal function. We have also performed additional experiments investigating the impact of GtACR2 activation on baseline astrocytic function, the results of which are in Supplementary Fig. 15. On the reviewer’s suggestion a statistical comparison of Fig. 2c and Fig. 2f has been carried out and is presented in Supplementary Fig. 7. No statistical differences between estimated DF_{GABAA} recorded using GABA puffs versus ORCHID could be detected. Unfortunately, such a comparison could not be performed in astrocytes as the Voltron2-ST construct is under the hSyn promoter, which is not active in astrocytes. The repeated activation data (Fig. 1l) includes recordings from both pyramidal neurons and interneurons.

4) Figure 3, the figure does not fit the stream of information presented in the results section. Different scales are used on all axis, which makes it difficult to compare the data. All shortcuts should be introduced in the text. Please introduce NHE. Figure 3j-m, has this approach been used and validated before? A reference is needed.

Although we acknowledge that the computational and theoretical data presented in Figure 3 represents something somewhat different to the rest of the experimental data, we believe it is both novel and important for delineating the biophysical basis of DF_{GABAA} , which we verify experimentally. Different time scales are used as the model has the advantage of tracking important variables over both extended timescales (Fig. 3b and d), together with shorter timescales which map more closely onto our experimental probing of DF_{GABAA} (Fig. 3c and e). We apologize for not including some abbreviations in the text, which we have now rectified both in the main text of the Results section and in the Fig. 3 legend. The approach used in Fig. 3j-m has been used and validated before (Düsterwald *et al.*, 2018), we now provide a reference in the text.

5) Figure 4, data from pyramidal and GAD2+ cells are shown, f-i are missing information which cell type this data is obtained from. In Fig. 4b, d: it is not obvious how these values are quantified; it is suggested that the corresponding periods (Baseline, Pre-SLE, Post-SLE, washout) be marked in the example traces (in 4a, c) and the process for quantification should be clearly described in the results and/or method section. The approach allows for a higher resolution than bulk averages, why are not the single recordings plotted over time?

For a description of the cell type from which the data in Fig. 4f-l were collected, please see the Fig. 4e legend, as well as the main text (“To this end, ORCHID was used to track DF_{GABAA} in the same GAD2+ interneurons before and after a 180 min treatment...”). We are grateful to the reviewer for mentioning the lack of clarity in the quantification method used in Fig. 4b,d (and Fig. 5d). We have now clearly labelled the corresponding periods as suggested in Fig 4a, Fig. 4c, and Fig. 5d (astrocytes), as well as Supplementary Fig. 9 (neurons) Supplementary Fig. 12 (astrocytes). Additionally, we now thoroughly describe the quantification method in the Methods section: “Here the averaging of multiple ORCHID

activations was not possible due to the rapid DF_{GABAA} changes. Instead, we took the mean signal during a single optogenetic activation (1 s) during each relevant period, which was used alongside an equivalent baseline period immediately prior to the optogenetic activation. The relevant periods were: < 15 s before SLE onset for pre-SLE, < 15 s before SLE cessation for during SLE (astrocytes only), and < 15 s after SLE cessation for post-SLE. For pre-SLE and post-SLE measurements, optogenetic activations where neither the activation nor the baseline period were contaminated by SLE-associated spiking or depolarization were selected (which excluded any measurements during GtACR2-triggered burst firing). For during SLE measurements in astrocytes, a measurement close to the end of the SLE, where measurements were typically more consistent and thus representative, was selected.” Finally, we agree that it is useful to plot the single recordings over time and have done so in Supplementary Fig. 9 (neurons) and Supplementary Fig. 12 (astrocytes).

6) Figure 4, and Figure 6 show DF_{GABAA} from GAD2+ recorded from slices or in vivo. While in slices DF_{GABAA} is always negative during baseline conditions, in vivo the baseline fluctuates around zero. Why are these baseline values so different? The same is true for values determined after 4-AP application.

We agree with the reviewer on this excellent point and have now added the following sentences to the discussion: “Our *in vivo* results captured a range of resting DF_{GABAA} values in cortical L1 GAD2+ interneurons, which is different to what was observed *in vitro*. We hypothesize that this heterogeneity may be due, in part, to differences in network state (Burman *et al.*, 2023) or the animal’s sleep-wake history (Alfonsa *et al.*, 2023).”

7) Figure 5, Figure 5d and e would benefit from more detailed analysis. It is clear in d that there is a huge efflux during the first blue light stimulus and that it becomes much smaller during the later stimulations. Please split the “during SLE” average shown in Figure 5e into first and second stimulus or plot single stimulations over time. Fig. 5d: How do the authors explain the strong initial increase in fluorescence at the start of SLE? This should be noted and discussed in the text.

We thank the reviewer for these points. Please see Supplementary Fig. 12 for the analysis of single stimulations over time. We have added an explanation to the text: “The large increase in fluorescence at the initiation of the SLE is likely due to a large depolarisation in the astrocytic V_m .” This is most likely in response to the high frequency of neuronal action potential firing that occurs initially at the start of the tonic period (with associated neuronal K^+ efflux). See the current clamp recording from the neuron (Fig. 5d).

8) Figure 5, activation of GtACR2 should be titrated in astrocytes as well. Furthermore, controls should be provided to characterize the impact of the optogenetic tool on astrocytic physiology. This is crucial to highlight the limitations of the tool. Fig. 5b: Did the authors expose astrocytes to longer duration of optogenetic activation? Does the depolarization continue or reach a plateau? Is the ORCHID method applicable to astrocytes if there is no unambiguous maximum?

We agree that the potential impact of the optogenetic tool on astrocyte physiology is important. Therefore, we measured astrocytic membrane potential and astrocytic membrane resistance (two fundamental readouts of astrocytic physiology) in our ORCHID expressing astrocytes in order to compare unexposed astrocytes to those who had received 3-5 mins of intermittent GtACR2 activation using blue light (the maximum duration we used in any of our

experiments). We found no detectable difference between the two groups. Please see Supplementary Fig. 15 for this new data.

For Fig. 5b and c, in order to enable a direct comparison, the duration of optogenetic activation used to estimate DF_{GABAA} for astrocytes was the same as that used for neurons (2 s). We agree that we may be underestimating DF_{GABAA} in astrocytes due to the fact that the glial membrane potential takes longer to change in response to opsin conductances which is a previously described phenomenon (Sasaki *et al.*, 2012), see Supplementary Fig. S3 C and D of Sasaki *et al.*, (2012). We now explicitly state this limitation in the Discussion. Because of this possible underestimation of DF_{GABAA} , our results in Fig. 5d use only within-cell comparisons.

9) The authors need to make clear in every section of the results, which preparation was used, brain slices or in vivo. Furthermore, the information about the number of animals, slices and cells should be included in all figure legends. All symbols should be explained, and it should be obvious in the figures which cell type has been recorded from. Please also provide information about the stimulation paradigm with the corresponding data. Specifically, figure 1f, what do the two different traces show? Is it both fluorescence, one representative trace and one averaged trace? Figure 3, g, i, m: do grey symbols represent single or (repeated) averaged measures? Fig. 5e: Again, what do the symbols represent? How are the individual phases defined?

We apologize for the lack of detail in places. The different sections of the Results and/or the figure legends now contain detailed information regarding the preparation used as well as the magnitude of n . We have also explained all symbols and have fully elucidated the cell types recorded from in each section of the Results. We have added information regarding the stimulation paradigm utilized and have fully explained Fig. 1f. We have added an explanation for the grey symbols in Fig. 3 (“Here and henceforth, grey data points indicate single recordings from individual cells, while black data points indicate the mean with error bars indicating SEM”). Finally, we have added an explanation for how the individual phases are defined in the Methods section and explained this more clearly in the relevant figures.

10) Immunohistochemistry is needed to confirm targeted expression of all constructs used.

We now provide evidence explaining our strategy for cell type specific targeted recordings and their validation using immunohistochemistry combined with other methods where appropriate. The following has now been added “Additional targeting and validation were performed to ensure recordings were made from the correct cell types. Cells that were CaMKII α + were confirmed as being pyramidal neurons by their morphology (large apical dendrite extending into stratum radiatum) and the location of their cell bodies in the stratum pyramidale of CA1/3. Further, patch-clamp recordings provided electrophysiological confirmation in a subset of cells (Supplementary Fig. 13). GAD2+ cells were confirmed as being interneurons by their morphology (multiple large processes extending from the cell body) and the occurrence of their cell bodies in stratum radiatum. Further electrophysiological characterization was used in a subset of cells (Supplementary Fig. 13). Additionally, immunohistochemistry was used to confirm that DIO-ORCHID and GAD1 were being co-expressed in the same cells in hippocampal organotypic slices from the GAD2-IRES-Cre mouse line (Supplementary Fig. 14, Methods). We confirmed the accurate targeting of astrocytes by GFAP-Cre through their morphological features (cells with small somata and multiple fine processes) and occurrence in the stratum radiatum. Additionally, patch-clamp recordings were performed from a subset of cells. In all cases ($n = 14/14$) we

found a low resting V_m (-72.25 ± 1.84 mV), low membrane resistance (59.76 ± 12.53 M Ω), and an inability to fire action potentials, which is typical of hippocampal astrocytes (O'Connor, Sontheimer and Ransom, 1994; McKhann, D'Ambrosio and Janigro, 1997) (Supplementary Fig. 15). We also confirmed that repeated stimulation of astrocytes expressing GtACR2 did not affect baseline electrophysiological properties (Supplementary Fig. 15)."

11) Page 17, line444-446, The authors claim that "detecting [Cl⁻]_i changes would require much larger and longer conductances that alter [Cl⁻]_i in the cytoplasm" this does not make sense regarding Cl imaging. Please clarify.

A clarification has been added: "Furthermore, as fluorescent reporters of [Cl⁻]_i are cytoplasmic, detecting [Cl⁻]_i changes would require much larger and longer conductances that alter [Cl⁻]_i in the cytoplasm, unlike ORCHID whose readout only requires Cl⁻ flux across the membrane to shift V_m ."

Minor:

1) Fig. 3 (line 228), what is gKCC2?

This abbreviation has been explained in main text and the Fig. 3 legend (gKCC2 is the conductance of KCC2).

2) The used constructs (Voltron-2 and GtACR2) should be referenced in the main text.

Voltron2-ST and GtACR2 are referenced in the introduction section of the main text.

3) The reference list needs to be checked carefully. Some citations are listed twice. References for the main text and method section should be merged.

We apologise for any errors in the reference list. We have checked it carefully and hope that these have all been corrected.

4) Please add Kaneko 2004 DOI:10.1523/JNEUROSCI.2115-04.2004 as reference for chloride FLIM. Page 4, line 48.

We agree that this is an appropriate reference and have added it accordingly.

5) Page 5, line 88-90, more information about the reason for using this dye is needed. It was mentioned later, but it causes confusion in the beginning of the result section. Please rearrange the text.

We thank the reviewer for pointing this out. The text has been modified for enhanced clarity: "Prior to imaging cells expressing ORCHID, a dye incubation step was carried out, with a dye that was selected for having spectral properties complementary to the experimental paradigm."

6) Page 7, line 151-152, Here the construct is called ORCHID even though it is the Voltron only. I recommend keeping a clear nomenclature for all constructs used, to help the reader follow the experiments.

We have now changed this wording in an effort to use more clear, consistent nomenclature as suggested.

7) Figure 2, please provide information why different wavelengths were used to excite Voltron.

JF₅₄₉ was used as spectral separation was not an issue when using GABA delivery to measure DF_{GABAA}.

8) Using the same scale on 2b and c, as on 2e and f would help to compare the collected data.

Please see Supplementary Fig. 7 for a direct comparison of the data collected using the two techniques using the same scale as requested.

9) Figure 2e, how long did the blue light stimulation last and why did you choose this duration?

The duration (2 s) has been added to the text where applicable, and the reason for choosing this duration based on targeted signal to noise ratios has been added to the Methods section (see Widefield fluorescence imaging).

10) There is a huge literature on NKCC1 in neurons and astrocytes, also transcriptomic data is available. Please include references and context based on previous publications.

We thank the reviewer for this recommendation. Citations to relevant studies have now been added.

11) Page 17, line 481, are able maintain, "to" is missing.

This has been corrected.

12) Methods, how were the mice killed?

The method of killing has now been added ("by cervical dislocation").

13) Please provide a reference for DIV7-14 being equivalent to P14-21

Please see the explanation in the Methods section. "Recordings were performed at 7-14 DIV, which is equivalent to P14-21 (as slices are prepared from P7 mice)."

References:

Abdelfattah, A.S. *et al.* (2023) 'Sensitivity optimization of a rhodopsin-based fluorescent voltage indicator', *Neuron*, 111(10), pp. 1547-1563.e9. Available at: <https://doi.org/10.1016/j.neuron.2023.03.009>.

Alfonsa, H. *et al.* (2023) 'Intracellular chloride regulation mediates local sleep pressure in the cortex', *Nature Neuroscience*, 26(1), pp. 64–78. Available at: <https://doi.org/10.1038/s41593-022-01214-2>.

Burman, R.J. *et al.* (2023) 'Active cortical networks promote shunting fast synaptic inhibition in vivo', *Neuron*, p. S089662732300586X. Available at: <https://doi.org/10.1016/j.neuron.2023.08.005>.

Cornejo, V.H., Ofer, N. and Yuste, R. (2022) 'Voltage compartmentalization in dendritic spines in vivo', *Science*, 375(6576), pp. 82–86. Available at: <https://doi.org/10.1126/science.abg0501>.

Düsterwald, K.M. *et al.* (2018) 'Biophysical models reveal the relative importance of transporter proteins and impermeant anions in chloride homeostasis.', *eLife*, 7, p. 216150. Available at: <https://doi.org/10.7554/eLife.39575>.

McKhann, G.M., D'Ambrosio, R. and Janigro, D. (1997) 'Heterogeneity of astrocyte resting membrane potentials and intercellular coupling revealed by whole-cell and gramicidin-perforated patch recordings from cultured neocortical and hippocampal slice astrocytes.', *The Journal of neuroscience : the official journal of the Society for Neuroscience*, 17(18), pp. 6850–63.

O'Connor, E., Sontheimer, H. and Ransom, B. (1994) 'Rat hippocampal astrocytes exhibit electrogenic sodium-bicarbonate co-transport', *Journal of Neurophysiology*, 72(6), pp. 2580–2589.

Sasaki, T. *et al.* (2012) 'Application of an optogenetic byway for perturbing neuronal activity via glial photostimulation.', *Proceedings of the National Academy of Sciences of the United States of America*, 109(50), pp. 20720–5. Available at: <https://doi.org/10.1073/pnas.1213458109>.

Wiegert, J.S. *et al.* (2017) 'Silencing Neurons: Tools, Applications, and Experimental Constraints', *Neuron*, 95(3), pp. 504–529. Available at: <https://doi.org/10.1016/j.neuron.2017.06.050>.

Reviewers' Comments:

Reviewer #1:

Remarks to the Author:

The authors have addressed my previous comments. Congratulations to the authors for this important study.

Reviewer #2:

Remarks to the Author:

In their manuscript entitled „All-optical reporting of inhibitory receptor driving force in the nervous system“ Joshua Selfe and coworkers described a new experimental approach to investigate one major aspect of GABAergic function, the driving force of the GABA_A-receptor (DF-GABA_A). As the DF-GABA_A is one essential parameter determining GABAergic inhibition and since a variety of studies described that the GABAergic driving force is influenced by a variety of physiological and pathophysiological conditions, the possibility to determine this parameter under minimal-invasive conditions with pre possibility of massive parallel recordings is advantageous. Therefore, the topic of this manuscript is highly relevant. The chosen approach is innovative and is adequately described, the results of the experiments are properly presented and illustrated. The discussion is fair, covers the advantages and limitations of this new method, and emphasized future applications of the ORCHID technique.

I appreciated the seriousness of the authors to respond to my comments and to alter the manuscript accordingly.

Therefore I strongly support the publication of this manuscript in Nature communications.

I only found three minor points, that may be corrected on the basis of the proofs:

Line 149 (Legend to Fig. 1) In my PDF the low-Cl⁻ datapoints were depicted in red and not in orange.

Line 435 (Legend to Fig. 6) It is not obvious to me whether the fragment “ – DF-GABA_A values plotted” is here on purpose.

Line 458: Please note that reference 1 and 2 are underlined.

Reviewer #3:

Remarks to the Author:

The authors did a great job improving the manuscript, controls were added, more clarity was added to the figures, the figure legends and the text. I have one major and a few minor additional comments:

major:

Thanks for adding a control testing the effect of GtACR2 in astrocytes. It is great that the tool does not change membrane voltage and resistance pre and post stimulus. I am still wondering whether the tool has side effects during stimulation. It has been shown before that optogenetic tools which affect Ca^{2+} in astrocytes result in K^{+} efflux, Cl^{-} tools affect K^{+} as well. How does the effect of GtACR2 activation compare to GABA puff mediated activation of GABAR?

minor:

Figure 3, please add the label GABAAR and GtACR2 to f and h, respectively. Similar to Figure 2.

Line 394, there is evidence that astrocytes express different amounts of functional NKCC1 in different brain regions. Please add the reference Engels et al.

10.3389/fncel.2021.735300 confirming NKCC1 impacting Cl^{-} in astrocytes in hippocampus.

References 15 and 56 are identical.

Line 26-27 “Intracellular Cl^{-} concentration ($[\text{Cl}^{-}]_i$) and hence EGABAA, is a dynamic variable that is known to differ between subcellular compartments, cell types, and over a range of timescales and network states⁴.” Please add a reference covering other cell types than neurons.

In line 371-373 you argue “expression to GFAP+ astrocytes in mouse hippocampal organotypic slices and optically estimated DFGABAA. We observed that, in stark contrast to neurons, the astrocytic DFGABAA under baseline conditions was strongly depolarizing, consistent with previous reports” The cited reference finds low Cl^{-} and argues that the depolarization is due to HCO_3^{-} currents. While the authors here argue mostly about Cl^{-} being the responsible anion. Please elaborate on this.

Response to reviewers

Reviewer #1 (Remarks to the Author):

The authors have addressed my previous comments. Congratulations to the authors for this important study.

We thank the Reviewer for the congratulations and endorsement of our work. There are no points to address.

Reviewer #2 (Remarks to the Author):

In their manuscript entitled „All-optical reporting of inhibitory receptor driving force in the nervous system “ Joshua Selfe and coworkers described a new experimental approach to investigate one major aspect of GABAergic function, the driving force of the GABA_A-receptor (DF-GABA_A). As the DF-GABA_A is one essential parameter determining GABAergic inhibition and since a variety of studies described that the GABAergic driving force is influenced by a variety of physiological and pathophysiological conditions, the possibility to determine this parameter under minimal-invasive conditions with pre possibility of massive parallel recordings is advantageous. Therefore, the topic of this manuscript is highly relevant. The chosen approach is innovative and is adequately described, the results of the experiments are properly presented and illustrated. The discussion is fair, covers the advantages and limitations of this new method, and emphasized future applications of the ORCHID technique.

I appreciated the seriousness of the authors to respond to my comments and to alter the manuscript accordingly.

Therefore I strongly support the publication of this manuscript in Nature communications.

We thank the Reviewer for their very positive description of our work and for providing their strong support for the publication of our manuscript.

I only found three minor points, that may be corrected on the basis of the proofs:

Line 149 (Legend to Fig. 1) In my PDF the low-CI- datapoints were depicted in red and not in orange.

We have improved the colouring in this Figure, so that it is now clearer that the datapoints are indeed orange.

Line 435 8Legend to Fig. 6) It is not obvious to me whether the fragment “ – DF-GABA_A values plotted” is here on purpose.

We thank the Reviewer for highlighting these small points and have now corrected them accordingly. The fragment “-DFGABA_A values plotted” is there on purpose, as well as in the legends of Figures 2 and 5. This is to clear up any possible misunderstanding between the plotting of -DF_{GABA_A} values in the figure, and the reporting of DF_{GABA_A} values in the figure legends.

Line 458: Please note that reference 1 and 2 are underlined.

This has been corrected.

Reviewer #3 (Remarks to the Author):

The authors did a great job improving the manuscript, controls were added, more clarity was added to the figures, the figure legends and the text.

We are delighted that that the Reviewer agrees we have done a great job in improving the manuscript. The Reviewer raises “one major and a few minor” additional comments, which we address below.

major:

Thanks for adding a control testing the effect of GtACR2 in astrocytes. It is great that the tool does not change membrane voltage and resistance pre and post stimulus. I am still wondering whether the tool has side effects during stimulation. It has been shown before that optogenetic tools which affect Ca²⁺ in astrocytes result in K⁺ efflux, Cl⁻ tools affect K⁺ as well. How does the effect of GtACR2 activation compare to GABA puff mediated activation of GABAR?

We thank the Reviewer for their comment regarding potential side effects of ORCHID upon ion concentrations in astrocytes. Of course any method that is used to estimate DF_{GABAA} will necessarily have to evoke a Cl⁻ flux, whether this is via optogenetic activation of GtACR2 or agonist activation of GABA_ARs (e.g. via a GABA puff). The potential side effects of these Cl⁻ fluxes upon ion concentrations (either upon Cl⁻ concentration directly, or indirectly upon K⁺ via K⁺/Cl⁻-linked cotransport, as alluded to by the Reviewer), will depend upon the magnitude and duration of the evoked Cl⁻ fluxes, regardless of the method used. Increasing the magnitude of the evoked conductance will increase the accuracy of the measured DF_{GABAA} , but will also increase the potential for side effects upon ion concentrations. The best way to minimise these side effects is to improve the experimenter’s temporal control of the duration and frequency of the evoked Cl⁻ conductances. In this sense, ORCHID offers an inherent advantage over an approach such as GABA puffing, as light stimuli typically offer the best temporal control - a point that we explain in our Discussion. However, some laboratories may have invested in their temporal control over GABA_AR activation by delivering an agonist via a piezo-based or uncaging approach, for example. For these reasons, we are not convinced that it would be interesting to determine how “GtACR2 activation compared to GABA puff mediated activation of GABA_AR” as the results will reflect the ability to control the evoked conductance sizes, which will vary across experimental setups.

With this in mind, we have instead conducted a new analysis on the stability of our estimates of astrocyte DF_{GABAA} using ORCHID, which is a way of assessing whether ORCHID is changing astrocyte ion concentrations under our experimental conditions. The data is included in a revised version of the manuscript (Supplementary Table 3; Methods), and shows that at the maximum frequency and duration at which we probed DF_{GABAA} using ORCHID in astrocytes, we did not observe changes in the DF_{GABAA} values measured within the same network state. For recordings made pre-SLE there was no statistical difference between DF_{GABAA} measurements made at least 20 s apart (paired t-test, two-tailed, $P =$

0.1756, $n = 10$ cells). Again, for recordings made post-SLE there was no statistical difference between DF_{GABAA} measurements made at least 20 s apart (paired t-test, two-tailed, $P = 0.6190$, $n = 8$ cells). These data support the conclusion that ORCHID is not causing significant changes to astrocyte ion concentrations under these experimental conditions.

minor:

Figure 3, please add the label GABAAR and GtACR2 to f and h, respectively. Similar to Figure 2.

This has been corrected.

Line 394, there is evidence that astrocytes express different amounts of functional NKCC1 in different brain regions. Please add the reference Engels et al. 10.3389/fncel.2021.735300 confirming NKCC1 impacting Cl in astrocytes in hippocampus.

We thank the Reviewer for this suggestion, and we have now added this pertinent reference to the manuscript.

References 15 and 56 are identical.

Our apologies, this has been corrected.

Line 26-27 "Intracellular Cl⁻ concentration ([Cl⁻]_i) and hence EGABAA, is a dynamic variable that is known to differ between subcellular compartments, cell types, and over a range of timescales and network states⁴." Please add a reference covering other cell types than neurons.

An additional reference covering glial cell types is now included.

In line 371-373 you argue "expression to GFAP+ astrocytes in mouse hippocampal organotypic slices and optically estimated DFGABAA. We observed that, in stark contrast to neurons, the astrocytic DFGABAA under baseline conditions was strongly depolarizing, consistent with previous reports" The cited reference finds low Cl and argues that the depolarization is due to HCO₃⁻ currents. While the authors here argue mostly about Cl being the responsible anion. Please elaborate on this.

The Reviewer is quite correct. It is for this reason that we were careful to refer to our driving force measurements made using ORCHID as DF_{GABAA} and not DF_{Cl} . This is because as we make clear in the manuscript "GtACR2's permeability sequence to different anions ($\text{NO}_3^- > \text{I}^- > \text{Br}^- > \text{Cl}^- > \text{F}^-$) is the same as other anion channels including GABA_ARs and GlyRs²⁶⁻²⁸, which are also permeable to HCO₃⁻". Therefore, we cannot exclude the possibility that differences in HCO₃⁻ currents or more specifically $DF_{\text{HCO}_3^-}$ differences between cell types or network states do not make important contributions to observed differences in DF_{GABAA} . To address this important point, the following sentence has now been added to the second paragraph of the Discussion: "As a result, it is possible that differences in intracellular pH and HCO₃⁻ resulting in $DF_{\text{HCO}_3^-}$ differences between cell types or network states could also contribute to observed differences in DF_{GABAA} made using ORCHID".

Reviewers' Comments:

Reviewer #3:

Remarks to the Author:

Thanks for addressing all my comments. I appreciate the additional work that has been put into this manuscript and I have no further comments.